



# Exploring mechanisms responsible for tidal modulation in flow of the Filchner-Ronne Ice Shelf

Sebastian H. R. Rosier [1] and G. Hilmar Gudmundsson [1]

[1]Department of Geography and Environmental Sciences, Northumbria University, Newcastle upon Tyne, NE1 8ST, UK

**Correspondence:** S. Rosier (sebastian.rosier@northumbria.ac.uk)

**Abstract.** An extensive network of GPS sites on the Filchner-Ronne Ice Shelf and adjoining ice streams show strong tidal modulation of horizontal ice flow at a range of frequencies. A particularly strong (horizontal) response is found at the fortnightly ($M_{sf}$) frequency. Since this tidal constituent is absent in the (vertical) tidal forcing, this observation implies the action of some nonlinear mechanism. Another striking aspect is the strong amplitude of the flow perturbation, causing a periodic reversal in the direction of ice shelf flow in some areas, and a 10-20% change in speed at grounding lines. No model has yet been able to reproduce the quantitative aspects of the observed tidal modulation on the Filchner-Ronne Ice Shelf. The cause of the tidal response has therefore remained an enigma, indicating a serious limitation in our current understanding of the mechanics of large-scale ice flow. A further limitation of previous studies is that they have all focused on isolated regions and interactions between different areas have, therefore, not been fully accounted for. Here, we conduct the first large-scale ice-flow modelling study to explore these processes using a viscoelastic rheology and realistic geometry of the entire Filchner-Ronne ice shelf, where the best observations of tidal response are available. We evaluate all the relevant mechanisms that have hitherto been put forward to explain how tides might affect ice-shelf flow and compare our results with observational data. We conclude that, while some are able to generate the correct general qualitative aspects of the tidally-induced perturbations in ice flow, most of these mechanisms must be ruled out as being the primary cause of the large observed nonlinear response. We find that only tidally-induced lateral migration of grounding lines can generate a sufficiently strong long-periodic $M_{sf}$ response on the ice shelf to match observations. Furthermore, we show that the observed short-periodic diurnal tidal motion, causing twice-daily flow reversals at the ice front, can be generated through a purely elastic response to basin-wide tidal perturbations in the ice shelf slope. This model also allows us to quantify the effect of tides on mean ice flow and we find that the Filchner-Ronne Ice Shelf flows on average ∼21% faster than it would in the absence of large ocean tides.

## 1 Introduction

Much of the Antarctic Ice Sheet is surrounded by ice shelves, and early research suggested that these ice shelves may have a significant mechanical impact on upstream flow (Thomas, 1973; Hughes, 1973). More recently, observations following ice shelf disintegration and modelling efforts have confirmed this 'buttressing' effect and enabled us to quantify it (Rott et al., 2002; De Angelis and Skvarca, 2003; Rignot et al., 2004; Furst et al., 2018; Reese et al., 2018). Ice shelves are now thought not only to modify upstream flow, but to have a potentially fundamental affect on the stability regime of grounding lines (Gudmundsson,





2013; Pegler, 2018; Haseloff and Sergienko, 2018). Understanding the mechanics of ice-shelf flow is therefore of considerable importance for assessing the future evolution of ice discharge from the ice sheet's interior, across grounding lines, and into the ocean.

Ice shelves are typically considered to flow at speeds determined by slow viscous processes. Recently, an increasing amount of GPS observations have revealed that the flow of both ice shelves and ice streams can be strongly modulated by ocean tides, leading to substantial temporal variations in velocity (Riedel et al., 1999; Doake et al., 2002; Legresy et al., 2004; Brunt et al., 2010; King et al., 2011; Alley, 1997; Bindschadler et al., 2003; Anandakrishnan et al., 2003; Gudmundsson, 2006; Marsh et al., 2013; Minchew et al., 2016; Rosier et al., 2017a). While the existence of tidal effects in ice-shelf flow is not surprising as such, these observations have revealed a number of unexpected and rather striking aspects of ice-shelf behaviour in response to a tidal forcing. Firstly, the horizontal response of the Filchner-Ronne ice shelf is strongest at a tidal frequency not found in the vertical tides that force it. As we discuss in more detail below, while the primary (vertical) ocean tidal constituents are the semi-diurnal ($M_2$, $S_2$) and the diurnal ($K_1$, $O_1$) ones, the single largest (horizontal) ice-shelf tidal constituent is the long-periodic $M_{\mathrm{sf}}$ tide with a period of 14.76 days (Rosier et al., 2017a). This $M_{\mathrm{sf}}$ constituent is absent in the vertical ocean tides beneath the ice shelf, an observation that implies the existence of some nonlinear mechanism capable of transferring tidal energy from short ($M_2$, $S_2$, $O_1$, $K_1$) to long ($M_{\mathrm{sf}}$) periods. Secondly, the tidal flow perturbations are much stronger and more widespread than one might have expected based on simple models of elastic flexure around grounding lines. The long-periodic response gives rise to 5 to 20 % changes in flow velocities across the whole ice shelf, with exact numbers dependent on location. Hence, these modulations in flow are not limited to the elastic flexure zone in the grounding zone. Finally, the shorter period tidal response can give rise to even larger changes in ice-shelf velocity due to their higher frequency. In some locations the tidal modulation is strong enough to cause a reversal in ice flow direction (Makinson et al., 2012).

The range of timescales over which tidal effects are seen to operate suggests that studying these processes may yield insights into both the elastic and viscous properties of ice, thereby providing an opportunity to constrain ice rheology over spatial and temporal scales of direct relevance to ice sheet models. Furthermore, the mechanical coupling between vertical ocean tides and ice flow occurs in the grounding zone, a particularly important and complex part of the ice sheet where our modelling efforts are often focused. Given both the complexity of this behaviour and the difficulties in reproducing it, modelling these processes can be viewed as a test of our understanding of how ice flows across grounding lines and through ice shelves to the ocean.

Previously, Brunt and MacAyeal (2014) studied the tidal response of the Ross Ice Shelf using an viscous model. While the assumption of viscous rheology appears justified for studies of secular ice flow, this assumption is not appropriate for a study of processes at tidal timescales where viscoelastic effects can be expected to play a significant role in the deformation of ice (Jellinek and Brill, 1956). In this paper, we use a full-Stokes 3D viscoelastic model to investigate the causes of the observed horizontal tidal modulation of ice flow over Filchner-Ronne Ice Shelf. Our particular focus will be to attempt to replicate both the strong semidiurnal modulation of ice shelf flow at the $M_2$ frequency together with the fortnightly modulation at the $M_{\mathrm{sf}}$ frequency whose origin is much debated. The model domain includes the entire Filchner-Ronne Ice Shelf and we test all the mechanisms (as summarised below) that have been suggested as responsible for generating the observed tidal response of the ice shelf.





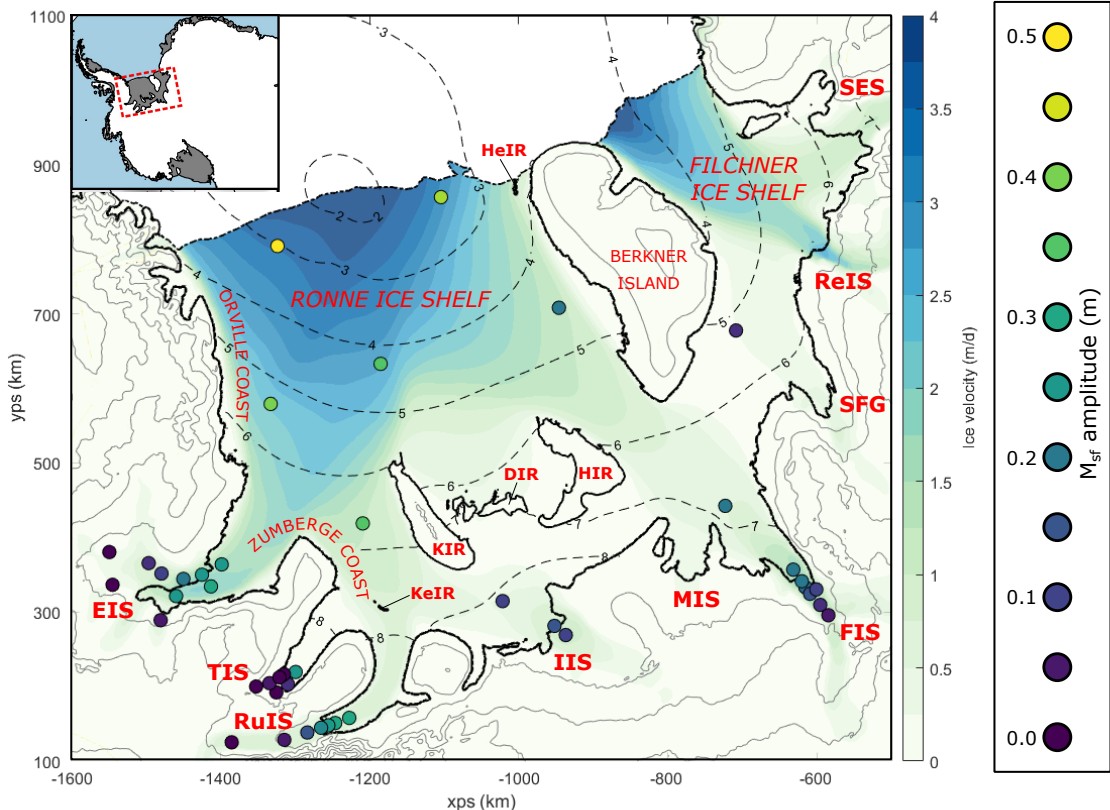

**Figure 1.** Map showing the Filchner-Ronne Ice Shelf and adjoining ice streams. Circular markers show locations of GPS sites and are coloured according to $M_{sf}$ amplitude. The background colourmap shows ice speed, grounding lines are indicated by solid black lines, dashed contour lines show tidal range and solid gray contour lines show ice surface elevation. Acronyms for glaciological features marked on the map are as follows: Evans Ice Stream (EIS), Talutis Ice Stream (TIS), Rutford Ice Stream (RuIS), Institute Ice Stream (IIS), Moller Stream (MIS), Foundation Ice Stream (FIS), Support Force Glacier (SFG), Recovery Ice Stream (ReIS), Slessor Ice Stream (SES), Hemmen Ice Rise (HeIR), Korff Ice Rise (KIR), Kershaw Ice Rumples (KeIR), Doake Ice Rumples (DIR) and Henry Ice Rise (HIR).

The paper is structured as follows: We begin by giving a background to ocean tides in the study area and an overview of observations showing tidal modulation of ice-shelf flow, followed by a summary of previous attempts at explaining these observations. We then present our numerical model and key results. Most of the detailed technical descriptions related the numerical experiments and implementations of different physical processes are found in several separate appendices. We find that many of the previously proposed explanations, while capable of producing a response at the correct frequencies, cannot generate a nonlinear response with a sufficiently large amplitude to match observations. Our proposed explanation, partly arrived by process of elimination, is that the tidal effects must to a large extent be caused by a tidally induced migration of grounding lines coupled with the nonlinear rheological response of ice.



## 1.1 Ocean tides beneath the Filchner-Ronne Ice Shelf

Tidal models constrained by GPS observations have lead to accurate knowledge of vertical tidal motion beneath the Filchner-Ronne Ice Shelf. Vertical tides at the grounding lines of ice streams feeding into the Ronne Ice Shelf are the largest in Antarctica, with a tidal range of up to 8 m (Fig. 1, dashed contours), making this an ideal location to investigate tidal effects on ice shelf flow.

Vertical tidal motion can be split into many tidal constituents, each with a unique period. Of these constituents, the semidiurnal $M_2$ (principal lunar) and $S_2$ (principal solar) constituents are the largest ones in this region. They are characterized by an amphidromic point, where their amplitude is zero and around which they rotate, located near the centre of the Ronne Ice Shelf front. These two constituents have periods of approximately 12.42 and 12 hours, respectively, and their combined wave envelope gives rise to the spring neap tidal cycle. This wave envelope should not be confused with the $M_{sf}$ frequency, despite having the same apparent 'period' of 14.76 days. The $M_{sf}$ frequency is found in spectral analysis of horizontal ice shelf and ice stream motion but not in the vertical tidal motion, since the semidiurnal wave envelope contains no energy at a fortnightly period. In addition to the important semidiurnal constituents, two diurnal constituents ($O_1$ and $K_1$) also have relatively large amplitudes, increasing near the grounding lines. Unlike the semidiurnal constituents these do not rotate around an amphidromic point and instead their phase increases approximately linearly through the region, from east to west.

## 1.2 Observations of tidally modulated horizontal ice-shelf motion

The first GPS observations to find tidal modulation of horizontal ice shelf flow were made in the Ekström Ice Shelf grounding zone (Riedel et al., 1999). In the years since, similar observations have been made on the Brunt (Doake et al., 2002; Gudmundsson et al., 2017), Ross (Brunt et al., 2010), Larsen C (King et al., 2011) and Filchner-Ronne ice shelves (Makinson et al., 2012; Rosier et al., 2017a) and the Mertz glacier tongue (Legresy et al., 2004). In terms of ice shelf velocities, different tidal constituents dominate depending on the local tidal regime. For example in the Ross sea the diurnal constituents ($O_1$ and $K_1$) dominate the vertical ocean tidal signal and these tidal constituents are also found most strongly in the (horizontal) ice shelf response. Conversely, in the Weddell Sea the semidiurnal constituents are strongest in both the vertical (Sec. 1.1) and horizontal motion (Makinson et al., 2012).

In this study, we focus our modelling efforts on the Filchner-Ronne Ice Shelf where we have a relatively large number of observations and the largest (vertical) ocean tidal amplitudes in the whole of Antarctica. A network of nine GPS stations made measurements spanning over a year in locations both near the Ronne ice front and further into the shelf (Makinson et al., 2012). These GPS measurements reveal strong horizontal tidal modulation at diurnal and semidiurnal frequencies. During the spring tide, at sites near the ice front, horizontal velocity and strain varied by ±300% of the mean values over a tidal cycle causing at times direction of flow to be reversed (Makinson et al., 2012). Further upstream from the calving front, horizontal diurnal and semidiurnal signals were found to decay to almost zero.

A more recent study that included several additional GPS sites near the outlet of major ice streams feeding the Ronne Ice Shelf found that the response over semidiurnal and dirunal periods increased again towards the grounding lines (Rosier



et al., 2017a). Further analysis of all available GPS data also revealed a strong $M_{\text{sf}}$ component in horizontal ice shelf motion across the entire floating ice shelf (Rosier et al., 2017a). This signal had previously been found on grounded ice streams (Gudmundsson, 2006; Minchew et al., 2016) and at a few locations on the Larsen C and Brunt Ice Shelves (King et al., 2011; Gudmundsson et al., 2017) but has now been found to occur over a vast area and with an amplitude typically greater than had

been found on ice streams (Rosier et al., 2017a).

### 1.3 Proposed mechanisms for tidal modulation of ice shelf flow

With the realisation that tides can strongly modulate ice flow, many different mechanisms have been put forward to explain these observations. Initially, Doake et al. (2002) suggested that currents beneath the Brunt Ice Shelf could be responsible for the tidal signals found there and this idea was explored further by Legresy et al. (2004) and Brunt and MacAyeal (2014). More

recently, the discovery of an $M_{\text{sf}}$ signal far upstream of the RuIS (Rutford Ice Stream) grounding line (Gudmundsson, 2006) has lead to a focus on replicating these observations (Gudmundsson, 2007, 2011; Rosier et al., 2014, 2015; Minchew et al., 2016; Robel et al., 2017) and much less has been done to understand the origin of horizontal tidal signals on ice shelves. Makinson et al. (2012) suggested that the large high frequency tidal modulation of the Ronne Ice Shelf could be explained by an elastic response to tilting of the ice shelf as tides rotate around the amphidromic point. Brunt and MacAyeal (2014) used a

purely viscous model of the Ross Ice Shelf to explore the effects of both tilt and currents but found that they could not replicate observations using their model. Neither of these studies investigated longer period modulation of ice shelf flow.

Efforts to understand the cause of the long period $M_{\text{sf}}$ signal, now known to occur across many of the large West Antarctic Ice Shelves where we have GPS data, require a nonlinear mechanism (Gudmundsson, 2007). The paucity of observations of this type on ice shelves led to an implicit assumption that any $M_{\text{sf}}$ signal observed on ice shelves was transmitted directly

from grounded ice rather than generated in floating regions. The increase in observations on ice shelves showing that the $M_{\text{sf}}$ signal often increases in amplitude downstream of grounding lines has made it clear that additional mechanisms must be at play (Minchew et al., 2016; Rosier et al., 2017a).

One possible nonlinear mechanism is that vertical tidal motion causes the grounding line to migrate back and forth sufficiently far as to have an effect on ice flow. Evidence of tidal migration of grounding lines remains relatively sparse but

measurements of this process have been inferred via remote sensing (Schmeltz et al., 2001; Brunt et al., 2011; Milillo et al., 2017) and cryoseismicity (Pirli et al., 2018). This could give rise to several nonlinearities: firstly this can result in the width of the ice shelf changing as the portion of floating ice changes over one tidal cycle (Minchew et al., 2016). In addition, the grounding line migration may be assymetric (Tsai and Gudmundsson, 2015) resulting in a greater migration upstream during high tide than downstream during low tide (Rosier et al., 2014). This idea of grounding line migration was explored in more

detail by Robel et al. (2017) who suggested that changes in buttressing arising from grounding line migration could explain observations on RuIS.

An alternative mechanism, termed "flexural ice softening", that results directly from the nonlinear rheology of ice itself, was put forward by Rosier and Gudmundsson (2018). The authors showed that tidal bending stresses in the grounding zone, which will vary in magnitude over a tidal cycle, could lead to a sufficiently large change in the effective viscosity of ice in this region





**Table 1.** Overview of the various model versions, including a brief description of how each one differs from the 'default setup' and the relevant appendix in which more details can be found.

| Model version name | Description | Appendix no. |
| --- | --- | --- |
| Default setup | The default model setup, as described in the methods | N/A |
| RF_n5 | Uses an exponent of 5 in Glen's flow law | Appendix A |
| RF_n4 | Uses and exponent of 4 in Glen's flow law | Appendix A |
| RF_Anoreg | Uses a rate factor of ice with no regularisation | Appendix A |
| RF_streams | Includes grounded ice streams in the domain | Appendix B |
| RF_damage | Includes damage effects in the ice rheology | Appendix C |
| RF_burgers | Uses a Burgers rheological model for ice | Appendix D |
| RF_GLmigration | Includes a parameterization of GL migration along model sidewalls | Appendix E |
| RF_currents | Includes sub ice shelf tidal current drag stresses | Appendix F |
| RF_temperature | Includes the effects of ice temperature on the rate factor of ice | Appendix G |
| RF_thinGZ | Reduces the thickness of ice in the grounding zone | Appendix H |

such that ice flow would be enhanced at high and low tide and lead to modulation of ice shelf flow at the $M_{\mathrm{sf}}$ frequency. Each of these nonlinear mechanisms could be playing a role in generating the observed $M_{\mathrm{sf}}$ signal.

Our main objective in this paper is to narrow down the possible causes of observed tidal motion of the Filchner-Ronne ice shelf by quantifying the contributions of several different processes and comparing these with observations. In particular we

5   will focus on reproducing the two strongest responses in horizontal ice-shelf flow: at the $M_2$ and $M_{\mathrm{sf}}$ frequencies.

## 2   Methods

In this section, we present a description of the 3D full-Stokes viscoelastic model used to investigate tidal modulation of ice shelf flow. What is described below is what we call the 'default setup' but several model experiments necessitated modifications or additional features to this setup. The various model versions that arise due to these differences are listed in Table 1 and each

10   is explained in more detail in the relevant appendix.

### 2.1   Field equations

We use a three-dimensional full-Stokes viscoelastic model in a Lagrangian frame of reference to solve for conservation of mass, linear momentum and angular momentum:

$$\frac{D\rho}{Dt} + \rho v_{i,i} = 0, \tag{1}$$



$$\sigma_{ij,j} + f_i = 0, \tag{2}$$

$$\sigma_{ij} - \sigma_{ji} = 0, \tag{3}$$

using the commercial finite element analysis software MSC.Marc MSC (2017). In the equations listed above, $D/Dt$ is the material time derivative, $v_i$ are the components of velocity, $\sigma_{ij}$ are the components of the stress tensor, $\rho$ is the ice density and $f_i$ are the components of the gravity force.

Ice rheology is represented by an upper-convected Maxwell rheological model that relates deviatoric stresses $\tau_{ij}$ and deviatoric strains $e_{ij}$ with

$$\dot{e}_{ij} = \frac{1}{2G}\overset{\triangledown}{\tau}_{ij} + A\tau_E^{n-1}\tau_{ij}, \tag{4}$$

where $A$ is the rate factor, $\tau_E = \sqrt{\tau_{ij}\tau_{ji}/2}$ is the effective stress, the superscript $\triangledown$ denotes the upper-convected time derivative, n is the constant in Glen's flow law, G is the shear modulus

$$G = \frac{E}{2(1+\nu)}, \tag{5}$$

$\nu$ is the Poisson's ratio and $E$ is the Young's Modulus (Shames and Cozzarelli, 1997). The rate factor is inverted for, as outlined in Appendix A, and the resulting surface ice velocity field is shown in Fig. 2c.

## 2.2 Boundary conditions

At the base of the ice shelf and along the ice front, a pressure is applied normal to the element faces given by

$$p = \begin{cases} \rho_w g(S(x,y,t) - z) & \text{if } z < 0 \\ 0 & \text{otherwise} \end{cases} \tag{6}$$

where $S(x,y,t)$ is the time varying local sea level (Sec. 2.3). At the ice surface a stress free boundary condition is imposed such that

$$\sigma \cdot \mathbf{n} = 0 \tag{7}$$

where $\mathbf{n}$ is the unit vector normal to the boundary. The only other boundary condition (BC) necessary in this 'default setup' is along the grounded sidewalls of the ice shelf domain. A Dirichlet BC is imposed on nodes along the grounded edge of the model, such that

$$u = u_{obs} \qquad\qquad v = v_{obs} \qquad\qquad w = 0 \tag{8}$$




where $u_{obs}$ and $v_{obs}$ are observed surface velocities (Rignot et al., 2017, 2011a; Mouginot et al., 2012). This BC allows for in-flow from fast moving ice streams but clamps nodes vertically at the grounding line such that ice in the grounding zone must bend to accommodate vertical tidal motion of the ice shelf. We use Bedmap2 (Fretwell et al., 2013) to define the model geometry; including ice thickness, ice surface elevation and grounding line position.

## 2.3 Tidal forcing

We use the circum-Antarctic inverse model (referred to hereafter as CATS2008a, which is an updated version of the inverse tidal model described by Padman et al. (2002)), to force our viscoelastic model at the ocean boundary with vertical tidal motion. By most measures, CATS2008a remains the best performing tidal model in this region (King et al., 2011). The Cats2008a model includes ten tidal constituents and we opt to force our model with the four largest constituents in the region by amplitude ($M_2$, $S_2$, $O_1$ and $K_1$), representing the principal semidiurnal and diurnal constituents. With these four constituents the most important tidal features are captured such as the spring neap cycle and the rotation of tides around Weddel Sea amphidromic point. Since the fortnightly $M_{sf}$ signal is produced through interaction between the $M_2$ and $S_2$ constituents then only these two are needed to investigate this phenomenon, however we add the diurnal constituents to ensure that the total tidal range is close to what is observed. The tidal model domain uses a slightly different grounding line to that used in the model presented herein, particularly at the outlet to Evans Ice Stream, so nearest neighbour interpolation is done to fill in areas with no amplitude or phase information in CATS2008a.

## 2.4 Element discretisation

The finite element mesh uses 3D, fifteen-node, isoparametric pentahedrons, arranged such that the triangular faces are oriented in the horizontal plane (Fig. 2b). The pentahedral shape is well suited to modelling an ice shelf in three dimensions, since the triangular faces enable the element to conform to a complicated coastline geometry without an excessive number of elements and the relatively flat surfaces of the ice shelf are captured well with the quadrilateral faces. The stiffness of the element is formed using 21-point Gaussian integration and triquadratic interpolation shape functions are used for displacements, resulting in a linear variation in stress through the element. The model mesh is unstructured and refined around grounding lines and in regions of high lateral shear strain (Fig. 2a). Certain model versions necessitate a different mesh but the total number of elements remains approximately the same and results in between $5 \times 10^5$ and $1 \times 10^6$ degrees of freedom.

## 3 Results

A number of different model setups were used in the course of this study. Our strategy, whose motivation will become clear later, was to begin by attempting to reproduce GPS observations of tidal motion at the principal semidiurnal ($M_2$) frequency. Once the model was able to reproduce these observations we subsequently searched for the source of the fortnightly ($M_{sf}$) frequency by testing a number of mechanisms and parameter combinations. Our results are therefore structured in that order and we



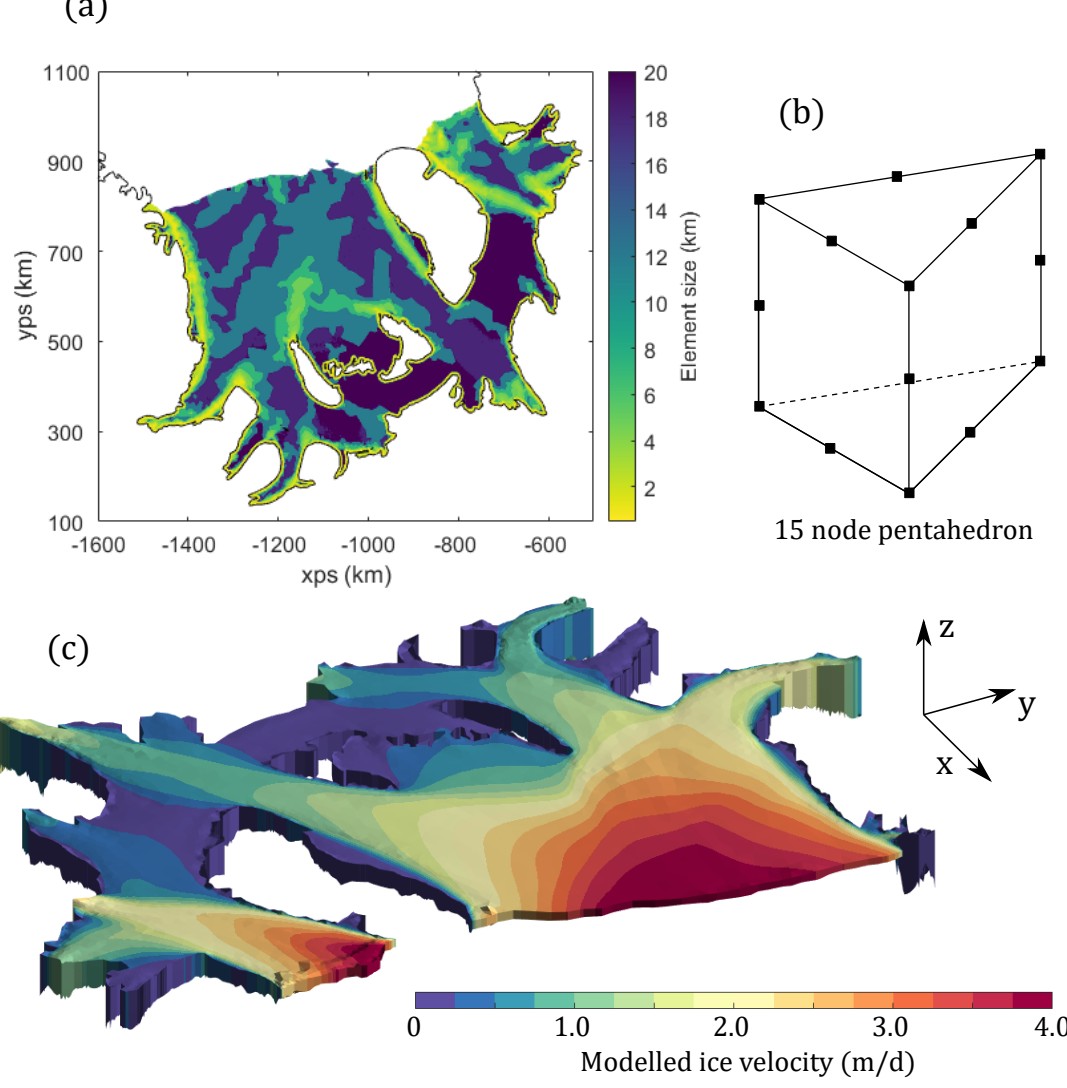

**Figure 2.** Overview of the finite element model, showing model resolution (panel a), the quadratic pentahedral elements used (panel b) and a vertically exaggerated oblique view of the model showing modelled mean ice velocity for the 'default setup' experiment.

begin by presenting model results at the $M_2$ frequency. In all cases, the amplitude of modelled horizontal tidal components was determined by processing surface nodal displacements using the Utide MATLAB package (Codiga, 2011).





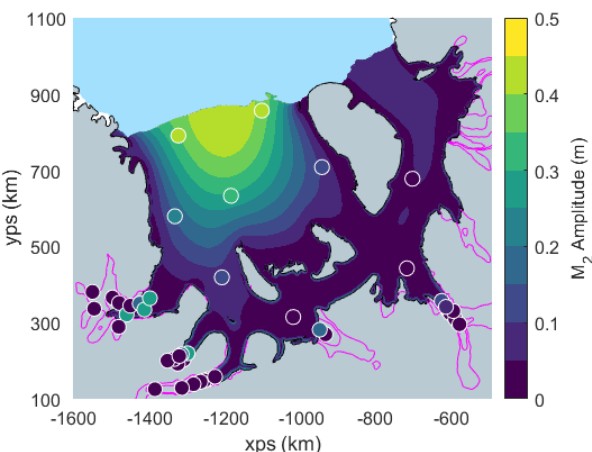

**Figure 3.** Modelled horizontal ice shelf motion at the $M_2$ frequency (background colourmap), compared with GPS observations (circles). Magenta lines are $0.5 \, \mathrm{m \, d^{-1}}$ ice velocity contours, to help identify regions of fast flowing grounded ice. Grounded ice is coloured gray, open ocean blue and ice shelves not included in the model domain are white.

## 3.1 Semidiurnal ($M_2$) results

Modelled horizontal ice shelf motion at the $M_2$ frequency only differed slightly between all the model experiments set out in Table 1 and so we only present the $M_2$ results from the 'default setup'. A large amplitude $M_2$ component in modelled ice shelf flow is found around most grounding lines and towards the ice shelf front (modelled $M_2$ amplitude is shown by the main

colourmap in Fig. 3). The maximum in the modelled $M_2$ signal is focused around the $M_2$ amphidromic point, near the centre of the ice shelf front, where the amplitude approaches 0.5 m.

    The difference between the modelled and observed response at the $M_2$ frequency is small over the majority of the ice shelf (GPS observations of $M_2$ amplitude are indicated by coloured circles in Fig. 3, using the same colour scale as the background colourmap). In particular, the bulk signal revealed by GPS observations of increasing amplitude towards the ice

front is reproduced by the model. The only discrepancy occurs at GPS locations located near grounding lines but outside of the immediate grounding zone. GPS observations at these locations reveal a stronger amplitude $M_2$ signal than that generated in the model, where it largely remains confined to the narrow grounding zone (Fig. 3). This mismatch is largest around the oulet of EIS, where modelled $M_2$ amplitude is $\sim$30% of the value measured by GPS. Any horizontal tidal motion must decay to zero at the grounding line as a result of our model BCs and so this mismatch is alleviated somewhat in the 'RF_streams' model

version where the $M_2$ amplitude is larger at the grounding line of major outlet glaciers.

    A number of model experiments were conducted specifically to better understand the model response at the $M_2$ frequency. Firstly, the Young's Modulus was both increased and decreased (between 1 and 9 GPa) to investigate how this affects the $M_2$ signal. This was found to only alter the amplitude of the $M_2$ effect and not it's spatial variability; an increase in $E$ reduced the

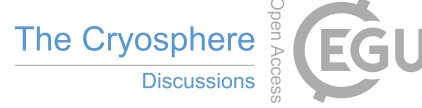

amplitude and vice versa for a decrease in $E$. Since the modelled $M_2$ amplitude is only sensitive to $E$, we treat it as a tuneable parameter and found that a value of $E = 2.4\,\text{GPa}$ produced a very good fit to observations.

Two further model experiments were carried out to confirm the mechanisms responsible for generating this $M_2$ signal in the model. In the first, the vertical Dirichlet boundary condition imposed along the sides of the ice shelf was removed, thus removing the effect of bending in the grounding zone. In this experiment, the bulk $M_2$ signal across the majority of the shelf was identical but no $M_2$ signal was generated in the grounding zone. The second experiment set the amplitude and phase of all tidal constituents to be the same in the whole domain (equal to the mean amplitude in this region). This stops the ice shelf from tilting due to either phase or amplitude differences across the domain. With this setup the model only generates an $M_2$ signal in the grounding zone and there is no horizontal motion at the $M_2$ frequency across the main bulk of the ice shelf.

## 3.2 Lunar synodic fortnightly ($M_{\text{sf}}$) results

A number of model experiments were conducted, including various different processes (as outlined in table 1) in an effort to determine possible sources of the observed fortnightly ($M_{\text{sf}}$) modulation in flow of the Filchner-Ronne Ice Shelf. We start by exploring the flexural softening mechanism proposed by Rosier and Gudmundsson (2018) and the modelled $M_{\text{sf}}$ amplitudes for some of those simulations are shown in Fig 4. The 'default_setup' experiment produces an $M_{\text{sf}}$ response whose spatial distribution appears to agree reasonably well with GPS measurements but the amplitude is much smaller than observed (Fig. 4a). Towards the ice shelf front, where the $M_{\text{sf}}$ signal has a measured amplitude of $\sim 0.5\,\text{m}$, the $M_{\text{sf}}$ amplitude in the model is an order of magnitude less. In and around the grounding zones, where a large number of GPS observations exist, the mismatch is equally bad.

The strength of the $M_{\text{sf}}$ signal generated through the flexural softening mechanism is sensitive to a number of different modelling choices and in particular to changes in the ice rheology since that is the source of nonlinearity. By increasing the exponent in the Glen-Steinemann flow law from $n = 3$ to $n = 4$ (Fig. 4b) or $n = 5$ (Fig. 4g, note the different colour scale used for this panel) the $M_{\text{sf}}$ amplitude can be increased substantially and its spatial distribution changes. For $n = 5$ the amplitude across the main portion of the shelf is approximately doubled but is still smaller than is measured by GPS. In these experiments with a higher exponent it becomes clear that modelled $M_{\text{sf}}$ amplitude largely originates from specific regions (e.g. the northern tip of KIR and along the Zumberge coastline) and spreads outwards onto the main shelf (Fig. 4g).

The rate factor of ice used in most experiments was inverted for using the Úa ice-flow model to solve the SSA momentum equations (Appendix A). As is typical in such inversions, some regularisation is imposed on the inverted field. We performed a suite of inversions using a range of values for the corresponding regularisation parameters. Since the rate factor not only determines mean ice velocity, but also the strength of the flexural softening mechanism, we also ran a model experiment ('RF_Anoreg') with no regularisation to see how a different spatial distribution in the rate factor (while resulting in the same mean ice shelf velocity) affect the calculated $M_{\text{sf}}$ amplitudes (Fig. 4f). The resulting $M_{\text{sf}}$ amplitude is generally larger than the 'default_setup', particularly along Western margins in the domain with a high spatial heterogeneity in the rate factor field where modelled $M_{\text{sf}}$ amplitudes are increased by $\sim 50\%$.





**Figure 4.** $M_{sf}$ amplitude on the Ronne Ice Shelf for nine model experiments outlined in table 1. Note the different colour scale used for experiment 'RF_n5' in panel g. Observed $M_{sf}$ amplitudes from GPS observations are indicated in panel a by coloured circles.

The 'RF_Burgers' experiment alters the rheological model from a Maxwell model to a Burgers model, thereby including the delayed elastic response not present in all other simulations (Appendix D). The additional rheological parameters were chosen along the lines of previous studies to replicate the observed $M_2$ amplitude and the resulting $M_{sf}$ amplitude was within 5-10% of the 'default_setup' (Fig. 4d). As with the 'RF_streams' experiment, this model version was computationally expensive and

5 since it had little effect on the modelled $M_{sf}$ amplitude the simpler Maxwell rheology was used for all other simulations.

Ice shelves are often heavily crevassed, affecting the behaviour of that portion of ice in response to a given stress. In terms of the viscous ice rheology this effect is already accounted for in the inversion for the ice rate factor, however its effect is not taken into account in the elastic response. To test how this might alter the $M_{sf}$ response we adopt a continuum damage mechanics approach whereby we calculate a damage field and reduce the Young's Modulus of ice in regions where damage is

10 high (as outlined in Appendix C). $M_{sf}$ amplitude of the 'RF_damage' model experiment is shown in Fig. 4i and is larger almost





everywhere in the domain than that calculated in the 'default_setup'. The change is greatest in and around narrow regions of ice shelf whose margins are calculated to be damaged but the $M_{\mathrm{sf}}$ amplitude is still almost an order of magnitude smaller than observations in these areas.

The geometry used in the model uses the assumption of hydrostatic equilibrium to calculate ice shelf thickness from known
surface elevation, but this assumption breaks down near grounding lines where bridging stresses become important. Given that flexural ice softening occurs in precisely this region and that it is sensitive to ice thickness (thinner ice increases the maximum elastic bending stresses but reduces the flexural length scale) this might cause its effects to be underestimated. There is currently no reliable method for estimating true ice thickness in the grounding zone and so instead we adopt a crude approach of reducing ice thickness in this region by a certain percentage everywhere in the model domain (Appendix H). The results of
this experiment, termed 'RF_thinGZ', are shown in Fig. 4e. The modelled $M_{\mathrm{sf}}$ amplitude is found to be relatively insensitive to the reduced ice thickness parameterisation that we employ, with amplitudes within 5% of the 'default_setup', ruling this out as a possible explanation for the large discrepancy between modelled and observed $M_{\mathrm{sf}}$ amplitudes.

Various authors have suggested that sub ice shelf tidal currents could play a role in modulating horizontal ice velocities at tidal frequencies (Doake et al., 2002; Legresy et al., 2004). We obtain tidal current amplitude and phase for the four largest
tidal constituents ($M_2$, $S_2$, $O_1$ and $K_1$) from the Cats2008 tidal model (Padman et al., 2002) and apply the resultant time varying drag to the base of the ice shelf in the 'RF_currents' experiment, as described in Appendix F. We ran the model with two different drag coefficients, a 'canonical' value of $3 \times 10^{-3}$ and an 'extreme' value of $3 \times 10^{-2}$. Modelled $M_{\mathrm{sf}}$ amplitude when using the higher drag coefficient are shown in Fig. 4h. These results reveal that strong tidal currents in combination with high drag coefficients could be locally important, increasing $M_{\mathrm{sf}}$ amplitudes in some areas by over 10% compared to the
'default_setup'. Overall, however, the effect of adding this mechanism is far too small to explain observations, even given an extreme value for the drag coefficient.

Most of our experiments do not include grounded ice in the model domain and instead we apply a Dirichlet boundary condition along the ice shelf grounding line. In order to investigate how this simplification affects the tidal response of our model, fast flowing grounded ice streams are included in the 'RF_streams' model version (Appendix B). Close to the grounding
lines of major ice streams this increases the $M_{\mathrm{sf}}$ amplitude, however over the bulk of the shelf the differences between the two simulations are generally less than 10% (Fig. 4c). Since including the grounded ice streams greatly increases the computational cost of running the model and we are focusing on replicating GPS observations across the main shelf we use this result to justify not including grounded ice in our domain.

We use the 'RF_GLmigration' model version (described in Appendix E) to run several experiments that explore the possibil-
ity that grounding line migration could be responsible for the pervasive $M_{\mathrm{sf}}$ signal observed on the Filchner-Ronne Ice Shelf. With our implementation of this mechanism, two parameters control the distance that the GL migrates up or downstream: $\gamma^+$ and $\gamma^-$, which determine the upstream and downstream migration distance respectively as a function of local tidal height. A simulation using $\gamma^+ = 6 \times 10^{-4}$ and $\gamma^- = \gamma^+/7.2$ yields a strong $M_{\mathrm{sf}}$ signal with an amplitude of $\sim 0.5\,\mathrm{m}$ near the ice front, matching reasonably well with observations (Fig. 5a). This choice of parameters is equivalent to a maximum upstream
migration of $5\,\mathrm{km}$ (or $\sim 700\,\mathrm{m}$ downstream) for a positive (or negative) vertical tidal motion of $3\,\mathrm{m}$.



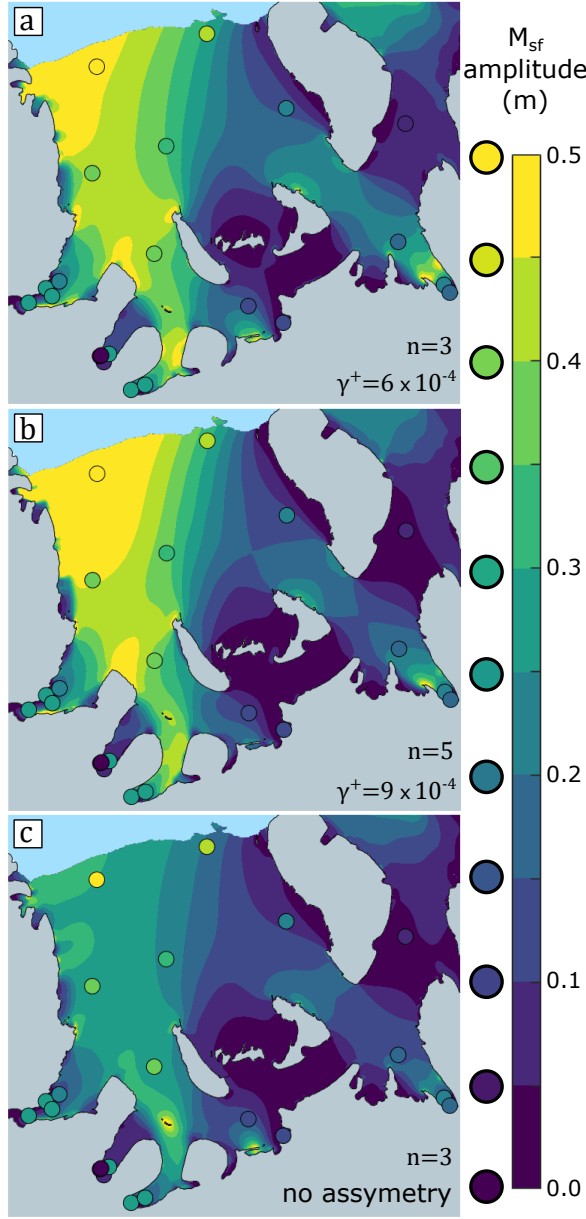

**Figure 5.** Modelled $M_{\mathrm{sf}}$ amplitude on the Ronne Ice Shelf for three variants of the RF_GLmigration model experiment. Panel a uses $n = 3$, $\gamma^+ = 6 \times 10^{-4}$ and $\gamma^- = \gamma^+/7.2$, panel b uses $n = 5$, $\gamma^+ = 9 \times 10^{-4}$ and $\gamma^- = \gamma^+/7.2$ and finally panel c uses $n = 3$ and $\gamma^- = \gamma^+$ (i.e. no assymetric grounding line migration). Observed $M_{\mathrm{sf}}$ amplitude is indicated by the filled circles using the same colour scale.

A second experiment was conducted using the 'RF_GLmigration' model version but with $n = 5$ in the Glen-Steinemann law to see how this affects the $M_{\mathrm{sf}}$ amplitude. Increasing the exponent in the flow law leads to a larger $M_{\mathrm{sf}}$ signal across the




shelf and so we can obtain an equally good match to the observed $M_{\mathrm{sf}}$ amplitude with a lower $\gamma^+$ of $9 \times 10^{-4}$ (Fig. 5b). In this case, upstream and downstream migration distances needed to fit observations are 33% smaller than with a choice of $n = 3$ but qualitatively the spatial pattern is very similar.

Tidally induced grounding line migration is generally expected to be assymetric between high and low tide (Tsai and Gudmundsson, 2015), however we can explore what happens to the $M_{\mathrm{sf}}$ amplitude if this part of the GL migration nonlinearity is removed (i.e. $\gamma^+ = \gamma^- = 1.05 \times 10^{-3}$). The resulting $M_{\mathrm{sf}}$ amplitude with a symmetric grounding line migration but the same total migration distance over one tidal cycle is shown in Fig. 5c. With symmetric GL migration the $M_{\mathrm{sf}}$ amplitude is generally $\sim$50% smaller, although still larger than experiments with no GL migration and certain areas such as the Kershaw Ice Rumples (KeIR) generate a strong $M_{\mathrm{sf}}$ signal. Overall, $M_{\mathrm{sf}}$ amplitudes are much closer to those observed by GPS measurements when

GL migration is asymmetric and a larger migration distance would be needed to match these observations if GL migration were symmetric.

## 4   Discussion

We find that strong modulation of ice shelf flow at the $M_2$ frequency is generated by tilting of the ice shelf, supporting the hypothesis of Makinson et al. (2012). This tilt occurs through a combination of phase differences in high tide times around

the domain and the lower amplitude vertical tides at the ice shelf front, leading to a rotating tilt vector centered around the $M_2$ amphidromic point in the Weddell Sea. The modelled $M_2$ horizontal ice shelf motion is very similar between all model experiments that used the same elastic rheological parameters and almost completely insensitive to changes in either the viscous ice rheology or the inclusion of other mechanisms such as tidal currents and GL migration. These results demonstrate that the short frequency modulation of ice shelf flow arises from simple linear elastic processes.

The modelled $M_2$ amplitude was only sensitive to our choice for the Young's Modulus ($E$) and a best fit to GPS observations was found using $E = 2.4\,\mathrm{GPa}$. This Young's Modulus can be thought of as an 'effective' value, relevant for tidal periods, since for a viscoelastic material the Young's Modulus is not constant but is a function of loading frequency. Linear elastic beam models have often been fitted to measurements of tidal flexure to obtain a wide variety of values for the 'effective' Young's Modulus of ice (Lingle et al., 1981; Stephenson, 1984; Kobarg, 1988; Smith, 1991; Vaughan, 1995; Schmeltz et al., 2002;

Sykes et al., 2009; Hulbe et al., 2016), however most of these mistakenly treat the resulting $E$ as a material constant and furthermore they are highly sensitive to local factors such as basal crevassing (Rosier et al., 2017b). Since the $M_2$ amplitude in our model is generated by the cumulative elastic strain across the entire shelf, rather than in one particular location (e.g. the grounding zone as in all cited studies), this model provides a much better opportunity to estimate the viscoelastic properties of ice shelves.

The amplitude of the modelled $M_{\mathrm{sf}}$ response was also found to be sensitive to the Young's Modulus, since this changes the bending stresses in the grounding zone. On the other hand, the strength of the short period $M_2$ signal was effectively independent to any changes in viscous ice rheological parameters and indeed almost any other change made to the viscous



model parameters. This finding justifies our strategy of first tuning the elastic properties of the model to match observed $M_2$ response before moving on to exploring the parameter space with respect to the $M_{sf}$ signal.

Our study investigates the two mechanisms that have been proposed to explain the generation of the $M_{sf}$ signal across the entire ice shelf: flexural ice softening and grounding line migration. The flexural ice softening mechanism generates an $M_{sf}$

signal as a direct result of the nonlinear rheology of glacial ice. Tidally induced grounding line migration can be subdivided into several nonlinear effects which will all play a potential role in generating an $M_{sf}$ signal: periodic narrowing and widening of the ice shelf, reduction in basal shear stress, reduced buttressing from ice rises/rumples and grounding line asymmetry. With our modelling approach we cannot address the reduction in basal shear stress which will occur as the grounding line migrates upstream on major outlet glaciers, however the other three have been tested.

While the flexural ice softening mechanism generates the general type or $M_{sf}$ response observed, we have found that the full extent of observations cannot be reproduced with any tested combination of parameters or additional model complexity. We, for example, explored the impact of varying the value of the stress exponent $n$ in the flow law, as well modifying the spatial distribution of the rate factor field. While both of these changes affected the modelled $M_{sf}$ amplitude, a significant and systematic differences between model and observed $M_{sf}$ amplitudes remained. The addition of basal drag from tidal currents,

damage effects, temperature effects, a thinner grounding zone and a more complex rheological model also did not significantly improve the fit to data. The flexural ice softening mechanism must be responsible for some of the observed $M_{sf}$ signal, since the nonlinear rheology of ice is well established, but it cannot explain the pervasive and strong $M_{sf}$ signal that is observed over the entire Ronne Ice Shelf. In conclusion, none of the above listed mechanisms, all of which *do not* include grounding-line migration, can replicate the observed spatial distribution of the $M_{sf}$ signal over the Filchner-Ronne ice shelf.

We find that the only mechanism capable of fully explaining the extensive large amplitude fortnightly modulation of RIS flow is grounding line migration. Despite the relatively simple implementation employed here, the addition of this process into the model can easily produce an $M_{sf}$ signal that matches observed $M_{sf}$ amplitude and the broad scale spatial distribution (Fig. 5a). It would presumably be possible to obtain an exact fit to observations by allowing the grounding zone geometry to vary spatially, however finding the optimal geometry is a daunting inverse problem beyond the scope of this paper.

A number of variants to the RF_GLmigration experiment were conducted to further explore this mechanism. Making grounding line migration symmetric (but ensuring that the total migration distance remains the same) causes a large overall reduction to the $M_{sf}$ amplitude, while increasing the amplitude in certain small areas such as around the KeIR and a grounded band near the outflow of Institute Ice Stream (IIS) (Fig. 5). Increasing the exponent $n$ in the ice rheology increases the $M_{sf}$ amplitude, or conversely reduces the distance that the grounding line needs to migrate to match the observed $M_{sf}$ signal by 33%. Without

better knowledge of the distance that grounding lines are migrating over the entire ice shelf this result cannot be directly used to estimate viscous ice rheology but obtaining these data is certainly possible (e.g. Schmeltz et al., 2001; Brunt et al., 2011; Milillo et al., 2017). The migration distances we investigate in this modelling study are within the range obtained by satellite estimates; in this region, Brunt et al. (2011) identified areas in which grounding lines were migrating almost 10 km over one tidal cycle.





Several features of the modelled $M_{sf}$ response for the RF_GLmigration experiment deserve specific mentions. Unsurprisingly, there is virtually no $M_{sf}$ signal generated in regions where the ice shelf flows slowly, such as behind Berkner Island and Henry Ice Rise. Somewhat more interesting is the lack of an $M_{sf}$ signal in our modelled results along the Eastern coast of the Ronne Ice Shelf, despite a very large $M_{sf}$ amplitude on the opposite Orville Coast (Fig. 5). This is because the shear margin

is located some distance from the grounding line (Fig. 1), meaning that migration of the grounding line is not felt by the main ice shelf (but where the grounding line and shear margin do coincide, at the Southern tip of Berkner Island, an $M_{sf}$ signal is generated). Upon closer inspection, many of the regions that generate a strong $M_{sf}$ signal coincide with areas where the shear margin is close to the grounding line.

Our implementation of tidally-induced grounding line migration is limited in several ways. Most importantly, without accu-
rate knowledge of bed geometry we have had to resort to assuming equal local bed slopes around all grounding lines. In reality, it is likely that in many areas the grounding line migrates only short distances (although conversely we may be underestimating the migration in certain areas). Furthermore, we do not allow the grounding line to migrate in regions of inflow to the ice shelf since our analytical approach would not yield accurate ice velocities across these grounding lines. However, as we are focused on matching the broad scale features of the $M_{sf}$ signal, and localised $M_{sf}$ generation is found to decay over relatively short
distances (Fig. 5c), these limitations are not expected to affect our main findings.

Several previous studies have shown that the nonlinear response to vertical tidal motion which generates the $M_{sf}$ signal also leads to a change in the mean ice velocity (Gudmundsson, 2007, 2011; Rosier et al., 2014; Rosier and Gudmundsson, 2018). Our regional model, that can broadly replicate the $M_{sf}$ signal across the Filchner-Ronne Ice Shelf, now allows us to quantify the magnitude of this effect on an ice shelf for the first time. We ran a repeat of the 'RF_GLmigration' experiment, but with no
vertical tidal forcing, and compared the mean velocity in this simulation with that of our best fit to the observed $M_{sf}$ signal. We find that, averaged across the entire ice shelf, ice flow is enhanced by ∼21% due to the presence of tides. Much of this tidal flow enhancement is confined to certain portions of the shelf where the local enhancement can be much higher, particularly along the Zumberge and Orville Coasts, i.e. ice flowing out from IIS, RuIS and Evans Ice Stream. This suggests that a potentially important feedback exists between tidal amplitude and ice shelf geometry, i.e. if the ice shelf were to thin or retreat it would
alter tides sufficiently to further compound changes in ice flow.

## 5   Conclusions

We are able to obtain a good agreement between our model and observations of short-periodic tidal modulation in horizontal flow. This study therefore allows us to confirm the previously untested theory of Makinson et al. (2012), that this behaviour arises from the linear elastic response of ice to changes in slope resulting from the relatively high frequency semidiurnal tides.
We are also able to replicate, here for the first time, the size and spatial pattern of the observed long-periodic (i.e. longer than a day) motion of the Filchner-Ronne ice shelf by including a nonlinear mechanism related to grounding-line migration over tidal cycles. As with other nonlinear mechanisms proposed previously, this involves a nonlinear energy transfer from the two main semi-diurnal tidal constituents ($M_2$ and $S_2$) to the long-periodic $M_{sf}$ tidal constituent.



Our physical modelling approach, and the availability of comprehensive data sets of spatial and temporal variability in ice flow across the ice shelf from remote sensing and numerous in-situ GPS sites, puts strong constraints on any proposed mechanism. Model parameters that, for example, affect the spatial pattern of secular ice flow (e.g. $A$ and $n$) also impact the modeled temporal variation in ice flow generated through tidal action. As a consequence, several hitherto potential mechanisms

for the generation of long-periodic tidal modulating in ice flow can now be be discounted as viable explanations.

A nonlinear response to tidal bending stresses has been previously proposed as a possible mechanism to explain observed long-period motion (Rosier and Gudmundsson, 2018) and in our model this does produce the same general type of a response. However, our large-scale modeling approach using realistic geometry of the Filchner-Ronne ice shelf shows that the generated long-period amplitude is too small and its spatial pattern inconsistent with observations. Hence, while this mechanism most

likely contributes to the observed signal, it plays a relatively minor role. We arrive at this conclusion after having conducted comprehensive parameter studies using different nonlinear viscoelastic rheological models (i.e. Maxwell and Burges models) and testing the effects of adding various processes including damage and temperature effects into the model. We therefore consider our modelling results, in this respect, to be robust.

We thus arrive at the lateral migration of grounding lines over tidal cycle as the most promising candidate for explaining the

observations of tidal modulation on Filchner-Ronne ice shelf. Our findings support previous work by Minchew et al. (2016) stressing the importance of periodic widening and narrowing of ice-shelves as a result of vertical tidal motion in modulating ice flow downstream of Rutford Ice Stream. In our model, the lateral migration of grounding lines gives rise to several different nonlinear mechanisms that all act together. First, the horizontal migration distance is an asymmetrical function of vertical tidal amplitude (this aspect of grounding line migration arises whenever the ice-thickness gradient changes across the grounding

line), and this is an example of a *geometrical* nonlinearity. Secondly, ice-shelf flow is a nonlinear function of width and stress for non-Newtonian fluid such as ice. This suggests that observations of tidal modulation of ice shelf flow can be utilized to extract information about rheology of ice. However, this can only be done if bed geometry and ice thickness across the grounding line are sufficiently well known to calculate the migration distance. Alternatively, if independent observations of lateral migration of grounding lines in response to tides are available, the migration distance can be prescribed directly, in

which case the stress exponent ($n$) can be solved for using a fairly simple modelling approach. Using tidal observations in this manner to extract information about ice rheology is an intriguing possibility that needs to be explored further.

Previous work on long-periodic tidal motion has focused on finding a mechanism that can, at least in principle, give rise to the surprising nonlinear tidal response observed, whereby the largest tidal amplitude in the horizontal motion is long-periodic and at a frequency not found in the forcing (i.e. in the vertical tides themselves). By now several such mechanisms have been

proposed and studied in considerable detail. Here, we have further narrowed the search for a viable mechanism by performing the first large-scale modelling of the whole of the Filchner-Ronne system. In doing so we have been able to reject some previously suggested potential explanations. Further observations, in particular of tidally induced grounding line migration, are required before any definitive statement about the primary cause of the tidal variation in ice flow can be made.



*Data availability.* The GPS data set that was analysed to determine horizontal tidal motion of the Ronne Ice Shelf is publicly available through the UK Polar Data Centre at http://doi.org/10.5285/4fe11286-0e53-4a03-854c-a79a44d1e356 and the analysis of that dataset is carried out in Rosier et al. (2017a).

## Appendix A:  Inversion for ice rate factor

In order to conduct our 3D tidal model experiments we need to ensure that the model reproduces the observed mean ice flow across the whole computational domain. This we achieve by inverting for the spatial distribution of the rate factor $A$ for any given value of the stress exponent $n$. Rather than conducting this inversion using our computationally demanding MSC Marc full-Stokes finite element model, we perform this inversion step using the vertically-integrated ice flow model Úa.

The Úa ice-flow model uses the finite element method to solve the shallow shelf (SSA) equations (e.g. Hutter, 1983; MacAyeal, 1989), as described in more detail in (Gudmundsson et al., 2012). The inversion procedure that we use in the Úa model seeks to minimise the misfit between observed and modelled surface velocities by minimising $\log A$, using the adjoint method to calculate the gradient of the cost function as first described for ice flow models by (MacAyeal, 1992, 1993). The cost function being minimised also contains a Tikhonov regularisation term penalising large spatial gradients in $\log A$. For all but one of our simulations we determine the amount of regularisation in our inverted field through an L-curve analysis. We also did experiments without any regularisation ('RF_Anoreg' experiment).

Using Úa the inversion procedure was conducted for a much larger domain than that of the full-Stokes model, including the entire drainage basin of the Filchner-Ronne Ice Shelf and for flow exponent values of $n = 3$, $n = 4$ and $n = 5$. The inversion algorithm was run for several hundred iterations until the change in cost function in each successive step was below some small tolerance value. The mean absolute difference between observed and modelled surface velocities after the completing the inversion (for the $n = 3$ inversion with regularisation) was $18.3 \, \mathrm{m\,a^{-1}}$ over the entire domain and $22.5 \, \mathrm{m\,a^{-1}}$ across just the ice shelf. Since the assumptions of the SSA, used by Úa, are well met for floating ice shelves we expect that using the resulting inverted fields of $A$ in our full-Stokes 3D model will produce almost an identical ice-flow velocity fields. We did, indeed, find this to be the case, see Fig. A1 showing a comparison between calculated and measured velocity fields as calculated with both the SSA model 'and full-Stokes model MSC Marc using the exact same $A$ distribution.

## Appendix B:  Modelling of grounded ice

The main challenge to address when adding grounded ice streams into the 'RF_streams' model domain is to reproduce the correct basal sliding velocity ($v_b$). We use a Weertman sliding law of the form

$$v_b = \beta \tau_b^m \tag{B1}$$

where $\beta$ is basal slipperiness, $\tau_b$ is the basal traction and $m$ is a stress exponent. Using an inverted slipperiness field from a shallow shelf model (as we can do for the rate factor on the ice shelf, see Sec. A) is not possible as the difference in the





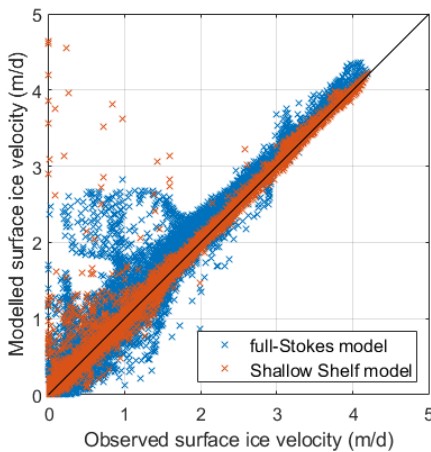

**Figure A1.** Comparison between observed surface ice velocities, obtained from the MEaSUREs version 2 dataset (Rignot et al., 2017, 2011b; Mouginot et al., 2012), and modelled ice velocities using either the 3D model (Section 2.1, blue crosses) or the Úa model (red crosses).

stress regime between the two models can become large on grounded ice and thus yields a velocity field that does not match observations.

   We use the inverse Robin approach outlined in Arthern and Gudmundsson (2010) that consists in solving the forward problem once with standard boundary conditions (denoted $\mathcal{NP}$) and then the slightly modified Dirichlet problem (denoted

$\mathcal{DP}$) in which the stress free upper surface condition (Eq. 7) is replaced by a Dirichlet condition where horizontal velocities derived from observations are imposed. The aim of this procedure is to minimize the Kohn and Vogelius cost function:

$$\mathcal{J}_{\mathrm{kv}} = \int_{\Gamma_b} \beta \left| \boldsymbol{v}^N - \boldsymbol{v}^D \right|_F^2 d\Gamma \tag{B2}$$

where $\boldsymbol{v}^N$ and $\boldsymbol{v}^D$ are basal velocities obtained by solving $\mathcal{NP}$ and $\mathcal{DP}$ respectively and $\left|.\right|_F^2$ denotes the Frobenius norm. The Gateaux derivative of the cost function $\mathcal{J}_{KV}$ with respect to the $A$ is given by

$$d_\beta \mathcal{J}_{\mathrm{kv}} = \int_{\Gamma_b} \beta' \left( \left| \boldsymbol{v}^D \right|_F^2 - \left| \boldsymbol{v}^N \right|_F^2 \right) d\Gamma \tag{B3}$$

   This derivative is strictly only exact for a linear ice rheology, nevertheless we use it here as an approximation to the true derivative as has been done by previous authors (Arthern and Gudmundsson, 2010; Gillet-Chaulet et al., 2012).

   Following Gillet-Chaulet et al. (2012), we make several modifications to the original approach described by Arthern and Gudmundsson (2010). Firstly we apply a change of variables and invert for $\alpha = \log_{10} \beta$ which avoids any un-physical negative

values appearing during the inversion procedure. This necessitates adding a correction to the gradient given in Eq. B3 of

$$d_\alpha \mathcal{J}_{\mathrm{kv}} = d_\beta \mathcal{J}_{\mathrm{kv}} \ln(10)\beta \tag{B4}$$





Secondly, because both data and model errors are present, it is necessary to avoid over-fitting the model to data which would lead to spurious changes in slipperiness. To this end, we impose a smoothness constraint on the total cost function by adding a Tikhonov style regularisation term of the form

$$\mathcal{J}_{reg} = \frac{\gamma}{2} \int_{\Omega} |\nabla \alpha|^2 \, dV \tag{B5}$$

5    where $\gamma$ is a parameter that must be carefully chosen to ensure a sensibile compromise between a smooth solution and small misfit. We perform an L-curve analysis to determine an optimum value of $\gamma = 10^{-3}$.

In the FEM context this regularisation term can be easily evaluated given the stiffness matrix, $[\mathbf{K}]$, such that the regularisation term in Eq. B5 is given by

$$\mathcal{J}_{reg} = \frac{\gamma}{2} \alpha^T \mathbf{K} \alpha. \tag{B6}$$

10    The gradient of this term with respect to $\alpha$ is simply

$$d_\alpha \mathcal{J}_{reg} = \gamma \mathbf{K} \alpha. \tag{B7}$$

Finally, the total cost function we seek to minimise is $\mathcal{J} = \mathcal{J}_{kv} + \mathcal{J}_{reg}$ with gradient $g = d_\alpha \mathcal{J}_{kv} + d_\alpha \mathcal{J}_{reg}$ given by Eqs. B4 and B7.

We use the native minimisation algorithm in MATLAB© to minimise $\mathcal{J}$. Once the inversion procedure has converged to 15    produce a slipperiness field for grounded ice we can use the sliding law in Eq. B1 as a boundary condition for the grounded parts of the model. The same Dirichlet BC that is used on grounded boundary nodes in the 'default setup' (Eq 8) is applied to nodes at the boundaries of the newly added ice streams.

## Appendix C: Calculation of a damage field

We employ a continuum damage mechanics approach to simulate the effects of damage on ice behaviour. In this framework, 20    the elastic modulus $E$ is pre-multiplied by a damage factor, $D$, such that $\tilde{E} = E(1 - D)$ where $\tilde{E}$ is now the effective elastic modulus of damaged ice. The damage factor can be thought of as a measure of the amount of voids in a given unit volume and a high damage factor reduces the portion of a body that can support a load. This simple approach means that in areas of high damage, such as where the ice is heavily crevassed, the effective elastic modulus is reduced and so the ice is less stiff.

We do not allow damage to evolve or advect on the shelf and instead we use the approach of Borstad et al. (2013) to calculate 25    a static damage field. The basic principal of their approach is to take the results of an inversion for the ice rigidity ($B = A^{1/n}$) and assume that spatial variations in this field arise from variations in temperature, back stress and damage. The damage field is then calculated as

$$D = 1 - \frac{B_{i-}}{B(T)}, \tag{C1}$$



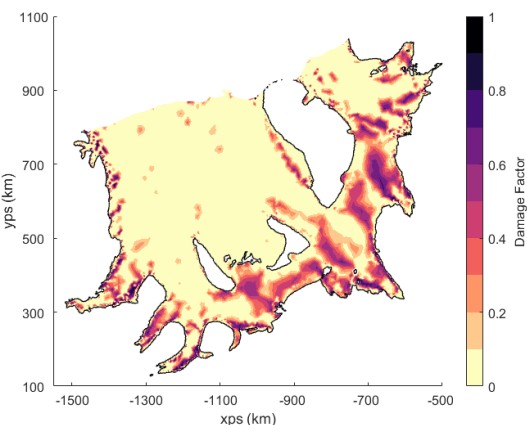

**Figure C1.** Damage factor field over the Filchner-Ronne Ice Shelf, as calculated with Eq. C1.

where

$$
B_{i-} =
\begin{cases}
B_i & \text{if } B_i \leq B(T) \\
B(T) & \text{otherwise}
\end{cases}
\tag{C2}
$$

and $B(T)$ is the ice rigidity that would be expected based purely on its dependence on temperature.

The only missing ingredient in the equations described above is a temperature field of the ice shelf in order to obtain $B(T)$. We use the analytical solution for ice shelf temperature from Holland and Jenkins (1999) which includes both diffusion and vertical advection. The solution assumes that vertical velocity in the ice shelf is equal to the basal melting rate ($w_B$) and that the ice shelf is in steady state, such that $w = (\rho_w/\rho)w_B$. In addition, surface and basal temperatures are needed as boundary conditions for the solution. We obtain the surface temperature field $T_S$ as a yearly average of the snow layer temperature from ERA-interim (Dee et al., 2011). We then fix temperature at the base of the ice shelf ($T_S$) at -2°C. The resulting vertical temperature profile is given by

$$
T(z) = \frac{(T_S - T_B)\exp\left(\frac{-wz}{\kappa}\right) + T_B - T_S \exp\left(\frac{wh}{\kappa}\right)}{1 - \exp\left(\frac{wh}{\kappa}\right)},
\tag{C3}
$$

where $\kappa = 1.14\text{x}10^{-6}$ is the ice shelf thermal conductivity. With this temperature field, we calculate the temperature dependant ice rigidity using the relation derived in Smith (1981).

The resulting damage field, as calculated from Eq. C1, is shown in Fig. C1. An exhaustive assesment of the resulting damage field is beyond the scope of this study, but areas of high damage are identified along shear margins giving some confidence that this calculation yields a sensible result. The damage factor enters the model by altering the modelled elastic behaviour of ice in regions where damage is high through a reduction in the effective elastic modulus, as outlined above.





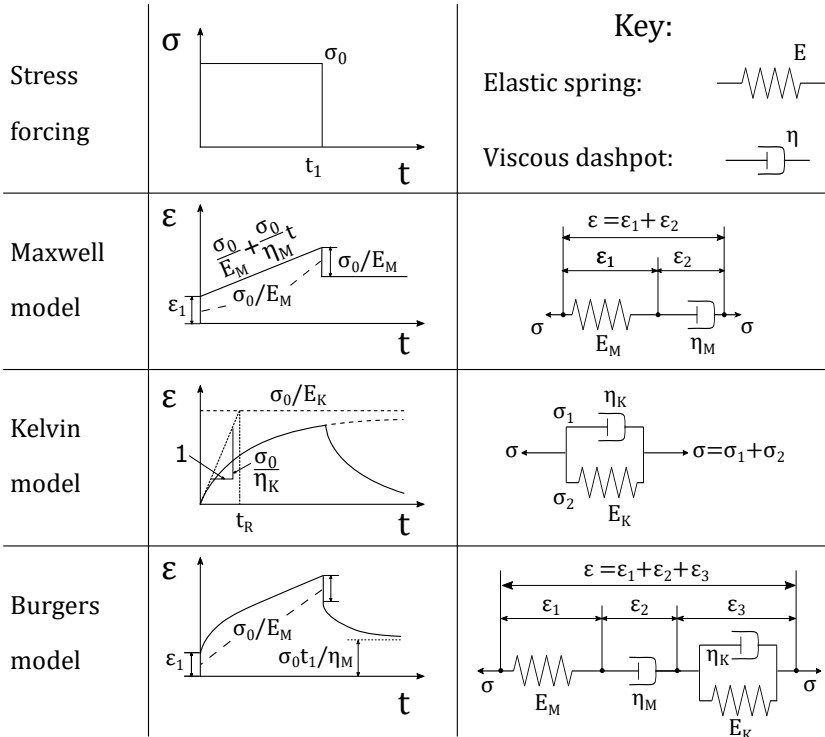

**Figure D1.** An overview of the three rheological models described in the text. The middle column shows a simple stress forcing initiated at $t = 0$ and removed at $t = t_1$ and the resulting strain response of the three models. Alongside the strain response of each model, in the far right column, is a schematic representation of each rheological model.

## Appendix D: Burgers rheology

The viscoelsatic rheological model used in the majority of full-Stokes ice-flow model experiments to date is the Maxwell model, consisting of a viscous dashpot and elastic spring connected in series (Sec. 2.1). When a stress is applied to a body of ice this rheological model captures both the instantaneous elastic deformation and the long-term viscous response and is one of

the simplest representations of viscoelastic behaviour. Another relatively simple rheological representation is given by a Kelvin model, consisting of a viscous dashpot and elastic spring connected in parallel. In this case the model exhibits a delayed elastic response commonly found in many viscoelastic materials and missed in the Maxwell model, but does not capture instantaneous elastic behaviour. A more complex model is needed to capture all of these behaviours and for that purpose we can use a Burgers model, consisting of a Kelvin and Maxwell element connected in series. Figure D1 shows a schematic representation of these

three rheological models and their behaviour for a stress applied at $t = 0$ and removed at $t = t_1$.



The constituitive equation of the Burgers model for deviatoric behaviour is given by

$$\tau_{ij} + p_1 \dot{\tau}_{ij} + p_2 \ddot{\tau}_{ij} = q_1 \dot{e}_{ij} + q_2 \ddot{e}_{ij}, \tag{D1}$$

where

$$p_1 = \frac{\eta_M}{G_M} + \frac{\eta_M}{G_K} + \frac{\eta_K}{G_K}, \tag{D2}$$

$$p_2 = \frac{\eta_K \eta_M}{G_K G_M}, \tag{D3}$$

$$q_1 = 2\eta_M \tag{D4}$$

and

$$q_2 = \frac{2\eta_K \eta_M}{G_K}. \tag{D5}$$

In the equations set out above, $\eta_M$ and $\eta_K$ are the viscosities of the Maxwell and Kelvin elements of the Burgers model, while $G_M$ and $G_K$ are the corresponding shear moduli (Shames and Cozzarelli, 1997). The volumetric deformation is assumed to be elastic and is defined in terms of the bulk modulus $K$ as

$$\sigma_{ii} = 3K\varepsilon_{ii}. \tag{D6}$$

The instantaneous elastic response of the Burgers model is determined by the shear Modulus $G_M$ in the Maxwell element and by the bulk modulus $K$. The Kelvin element parameters, $G_K$ and $\eta_K$, determine the delayed elastic response (primary creep). Finally, the steady-state viscous strain which is the only behaviour included in most ice shelf models is determined by the viscosity of the dashpot in the Maxwell element ($\eta_M$). Following Gudmundsson (2011) and Reeh et al. (2003), we use the values $G_M = 3.5$GPa, $K = 8.9$GPa, $G_K = 3.3$GPa, $\eta_K = 600$ GPa s and set $\eta_M$ to the standard viscosity used in Glen's flow
law. A major disadvantage of using a Burgers rheology is that, due to the short relaxation time of the Kelvin element (of the order of minutes), the model requires very short time steps to accurately model the primary creep behaviour.

## Appendix E:  Grounding line migration

Calculating tidally induced grounding line migration directly, by solving the contact problem, has been done previously using this model in a 2D flowline case (Rosier et al., 2014). Solving the contact problem for a high resolution model of the entire
Filchner-Ronne region would be prohibitively expensive computationally. Furthermore, with no detailed information on the bed slope around (and particularly downstream of) grounding lines this approach would still not yield a reliable assessment of grounding line migration. For these reasons, we reject this methodology and instead we adopt a simpler approach using





the analytical solution for grounding line migration proposed by Tsai and Gudmundsson (2015), which gives upstream and downstream migration distances as $\Delta L^+ = \Delta h^+/\gamma^+$ and $\Delta L^- = \Delta h^-/\gamma^-$, where

$$\gamma^+ = \beta + \frac{\rho}{\rho_w}(\alpha - \beta), \tag{E1}$$

$$\gamma^- = \frac{\gamma^+}{(1 - \rho/\rho_w)} \tag{E2}$$

and $\Delta h^{+/-}$ is a positive or negative vertical tidal motion. These equations assume hydrostatic equilibrium and constant bed ($\beta$) and surface ($\alpha$) slopes. An important result is that, under these assumptions, the upstream migration distance will be greater than the downstream migration distance for a positive or negative tidal deflection of the same amplitude. This analytical solution agreed reasonably well with GL migration calculated by solving the contact problem in the flowline version of this
model (Rosier et al., 2014).

Since, as stated above, we have no accurate knowledge of bed slopes in the model domain we assume that for the purposes of grounding line migration slopes are the same everywhere. This means that the distance the grounding line migrates in the model becomes a function of local tidal amplitude and phase only.

Implementing this into the model involves several steps. Firstly we calculate the distance of every node from the default
grounding line position as given by Bedmap2, $d_{GL}$, where distance is defined as positive for nodes upstream of the grounding line and negative for nodes downstream. Then, given the current local sea surface height, we calculate the current grounding line position using the equations derived by Tsai and Gudmundsson (2015). The basal boundary condition is then applied to nodes along the base of the ice shelf as follows:

$$u, v, w = \begin{cases} 0 & \text{if } d_{GL} \geq \Delta h \\ \text{free} & \text{otherwise} \end{cases} \tag{E3}$$

A weakness of this approach is that it cannot be used in a region where ice is flowing rapidly across the grounding line, ie. at the outlet of major ice streams, since the basal sliding velocity of grounded nodes is set to zero. Therefore we do not apply this BC in these locations.

A different finite element mesh is needed for any experiments that include GL migration, since the domain must include a band upstream of the GL which it can migrate into. We add a 10 km band of ice everywhere upstream of GLs except those at the
outlet of major ice streams. Mesh resolution in this band is $\sim 500$ m to resolve GL motion as well as possible without creating an excessively large number of additional elements. A schematic showing how grounding line migration is implemented in our finite element model is shown in Fig. E1.

**Appendix F: Sub ice shelf currents**

In order to simulate the effect of tidal current drag on the ice shelf, we use the depth averaged tidal currents from the Cats2008a
model (Padman et al., 2002). This provides amplitude and phase of the $U$ and $V$ components of tidal current transport for




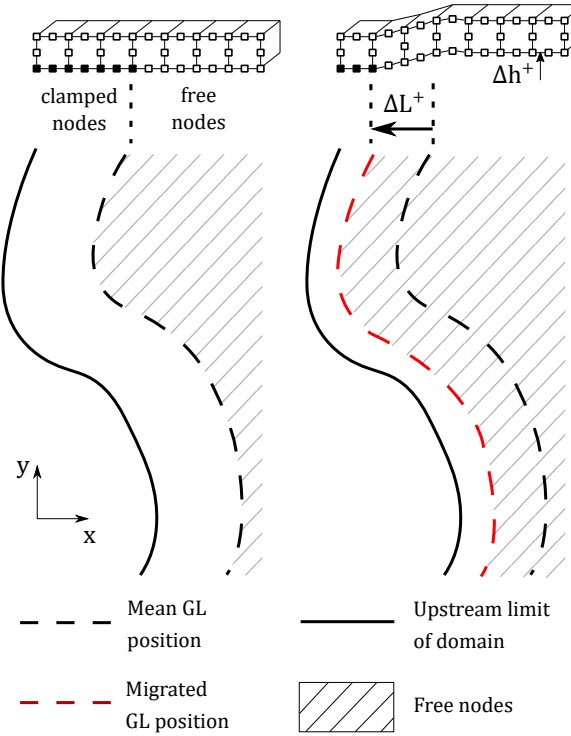

**Figure E1.** Plan view of a grounding zone, showing how grounding line motion is imposed in the model. Initially the model runs with no tides until it has viscoelastically relaxed. Nodes upstream of the grounding line are clamped at the base and nodes downstream have no constraints. Once tides are switched on, the grounding line can migrate upstream ($\Delta L^+$) or downstream ($\Delta L^-$) and nodes that outside of the newly defined grounded area have any constraints removed.

each of the four major tidal constituents beneath the shelf ($M_2$, $S_2$, $O_1$ and $K_1$). Following Robertson et al. (1985); Brunt and MacAyeal (2014), the $x$ and $y$ components of basal shear stress ($\tau_x$ and $\tau_y$, respectively) resulting from tidal currents are defined as

$$\tau_x = \frac{C_D \rho_w U \left| \mathbf{U_T} \right|}{H^2}, \qquad\qquad \tau_y = \frac{C_D \rho_w V \left| \mathbf{U_T} \right|}{H^2} \qquad\qquad (\text{F1})$$

5     where $H$ is water column thickness, $C_D$ is the drag coefficient and $\mathbf{U_T} = \sqrt{U^2 + V^2}$ is the magnitude of the tidal current transport vector. The drag coefficient is an unknown related to the roughness of the ice shelf base. We investigated two choices for $C_D$, a typical value of $0.003$ (Robertson et al., 1985; Brunt and MacAyeal, 2014) and a much higher value of $0.03$ which we use to test a theoretical upper limit on tidal current effects. The basal shear stresses are applied directly the bottom face of the ice shelf elements at the same time as vertical tides are initiated within the model.





## Appendix G: Temperature effects

The effects of temperature on ice rheology are, to a certain extent, included in every version of the model; since a part of the spatial variation in the inverted rate factor field is caused by temperature. One aspect of temperature variations that is not included however is the change in temperature with depth. This will result in ice that deforms more easily at the base than at

the surface, since ice will generally be cooler at the surface and warmer at the ice shelf base. This could conceivable play an important role in the response of an ice shelf to tides and is particularly relevant for the tidal flexure mechanism since here it is the ice rheology that is the source of nonlinearity that we seek to reproduce.

We are specifically interested in the effects of depth variation in ice softness and to explore this in the simplest way possible we assume that, everywhere in the ice shelf, temperature varies linearly from -20°C at the surface to -2°C at the base. We then

make the rate factor a function of temperature using the relation derived by Smith (1981).

## Appendix H: Thin grounding zone experiment

Ice thickness within the grounding zone (GZ) is modified in the 'RF_thinGZ' experiment to explore how this might affect generation of the $M_{sf}$ signal, in particular by changing flexural stresses resulting from tidal bending. The crude approach that we take for a first order estimate of this affect is to multiply ice thickness $h$ by a factor $1 - \lambda$ that is a function of distance from

the grounding line ($d_{GL}$), defined such that at the grounding line and far away from the grounding zone $\lambda = 0$ but within the grounding zone $0 < \lambda < 1$. The arbitrary function that we choose for this purpose is a log-normal distribution centered around $d_{GL} =$1km, where it reaches a maximum value of 0.1, i.e. leading a maximum reduction in ice thickness of 10%. Multiplying ice thickness by this factor yields a new ice thickness field that retains its overall shape and the effect is confined such that $\lambda \approx 1 \times 10^{-2}$ at $d_{GL} =$10 km and $\lambda \approx 1 \times 10^{-3}$ at $d_{GL} =$20 km (beyond this distance from the grounding line we do not alter

ice thickness).

*Acknowledgements.* We are very grateful to Andrew Bell for his invaluable help with MSC Marc. Sebastian H. R. Rosier was funded by the UK Natural Environment Research Council large grant "Ice shelves in a warming world: Filchner Ice Shelf System" (NE/L013770/1).





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
