# Peer review of "Exploring mechanisms responsible for tidal modulation in flow of the Filchner-Ronne Ice Shelf"

_The Cryosphere, 2019_

## Referee Comment (RC1) · Julia Christmann (Referee) · 18 Jun 2019

The paper "Exploring mechanisms responsible for tidal modulation in flow of the Filchner-Ronne Ice Shelf" by Sebastian H. R. Rosier and G. Hilmar Gudmundsson presents an interesting comparison of GPS measurements and simulation results considering the influence of ocean tidal forcing. The authors explore the impact of different mechanisms on tidal modulation in horizontal flow. First, the modelled semi-diurnal tidal constituent $M_2$ for the whole Filchner-Ronne Ice Shelf is compared to results of GPS measurements and afterwards the same is done for the long-periodic tidal constituent $M_{sf}$. The authors claim that the relatively high amplitude for $M_{sf}$ can only be obtained if the model allows grounding line migration.

General comments:
I have a number of major concerns with this paper, which are detailed below. In addition, some more specific comments and technical corrections are listed at the end of this review.

- The authors show in Table 1 an overview of the various model versions. Can you also include a Table where all parameters of the default setup are included, please? Then it is directly clear to the reader of the paper which parameter change is done for a different model setup and all parameter values are defined. For example, a value for Poisson's ratio is completely missing in the paper.
- For most model setups (except RF_streams), the velocity directly at the grounding line is set to an observed value but assumed to be constant – i.e. independent of the tidal forcing. In reality, the tides will have a big influence on the velocities near the grounding line, which is also verified by GPS measurements the authors show in this paper (shortly noticed p.10, l.13-15). The major simplification of a constant horizontal velocity is hardly discussed in the paper. Would the results be different if the horizontal velocity at the GL depends on tidal forcing? This could be checked by adding a sinusoidal behaviour onto the prescribed velocities at the boundary, with amplitudes and frequency derived from measurements. In Figure 4c (RF_streams), several ice streams are added but the results are only slightly better. However, how long are the added ice streams? At their boundary, the authors also prescribe constant velocities. So maybe it is impossible to model the influence of the tides by this extended geometry as the length of the grounded ice parts is too short to get independent horizontal velocities (of the inflow boundary condition) at the GL. Why can the authors do this major simplification and then state by the process of elimination that the strong long-periodic $M_{sf}$ component is only obtained by grounding line migration?
- Figure 2 (c): Why are there such large excursions in the geometry of the Filchner-Ronne Ice Shelf? For example directly at the ice shelf front of Filchner Ice Shelf or on the left and right side of Ronne Ice Shelf. Also at other places irregularities are noticeable.
  In the caption, the authors forgot to include (panel c).
- To better understand the effects of the various model setups shown in this paper, it would be nice to have relative deviations between observed and modelled results. For instance, in Fig. 4 the reader can imagine the influence of the different cases, but cannot really detect how a certain change in the setup reduces or increaes the difference between simulation and observation. The Filchner-Ronne Ice Shelf covers a large area and the GPS measurements are only points increased in size to be visible.
- Did the authors do a study considering the influence of the applied mesh to the results? From my experience, a Maxwell model as every viscoelastic material model needs a very very fine mesh resolution (I agree that it is not possible to get such a fine mesh resolution for the whole Filchner Ronne Ice Shelf) to generate reasonable results that are nearly mesh-independent. Is the mesh resolution the authors take not too coarse for a viscoelastic material model? Are the results reliable?
  An additional question about the mesh that remains unclear for the reader: How many layers are used for the vertical model coordinate?
- p.15, l.20-29: Why is Young's Modulus not a material constant? In all tested setups in your paper Young's modulus is constant using E=2.4 GPa except in the damage setup where Young's modulus is spatially changed but not in time. Or did I understand this incorrectly? Why should Young's modulus be a function of loading frequency for a viscoelastic material? I also wonder why are the viscoelastic

properties of ice shelves better provided by your simulation? And the last point I do not really understand is what do you mean by cumulative elastic strain, if you fit Young's modulus to GPS observations?

- p.16, l.6-9: In which setup did the authors test periodic narrowing and widening of the ice shelf and reduced buttressing from ice rises/rumples?
- In my opinion, the conclusion would benefit from a reduction to clearly stated important take-home messages. Maybe the authors can move some of the statements from the conclusions into the discussion?

Specific comments and questions:
- p.2, l.15,16: Give a reference. What is the expectation based on simple models of elastic flexure? Are the numbers in the next line results the authors show later on?
- p.4, l.32: How long are the measurements of the additional GPS sites near the outlet glaciers?
- Table 1: Include an additional table that contains the material parameters for the default setup. Then it is clear which changes between the various model versions are done. What is the value of Poisson's ratio?
- What are the unknows of the model (velocities or displacements), i.e. do the authors use a velocity or displacement formulation for the viscoelastic material model? For the boundary conditions, the authors use Dirichlet conditions for the velocities (p.7, l.25) and also show the resulting ice velocity field in Fig. 2c, but for the element discretization the authors write (p.8, l.22) "triquadratic interpolation shape functions are used for displacements". Is an arbitrary Lagrangian–Eulerian moving grid included in the model or how are the surface nodal displacements (p.9, l.2) determined in the model?
- p.7, l.18: For the water pressure applied at the ice front one needs the freeboard also depending on the time-varying local sea level (see the if condition "z<0"). Is this only a typo or how do the authors implement this?
- p.8, l.2-12: The reader cannot see the increase of the $M_2$ amplitude towards the grounding zone. A second figure with a magnification near a chosen grounding line would be helpful.
- p.11, l.4-6: When the vertical boundary condition is removed, the grounding line has to move and in my view, bending will always occur near the real position of the grounding line but maybe not at the position where the grounding line was before. What do the authors mean with "removing the effect of bending in the grounding zone"?
- p.13, l.35: What happens for a positive or negative tidal motion of 4 m, which fits better to the tidal range given in Fig. 1 for the grounding line region?
- p.21, l.7: Why did the authors put square brackets around K?
- p.22, Fig. C1: The damage factor could reach a value of 0.8 and below in interesting regions, for example at the boundary of inflow regions to the ice shelf. In the text, the authors stated values of E between 1 and 9 GPa, but 0.2*2.4 GPa = 0.48 GPa. Are these realistic (meaning physically useful) values? In my opinion, E has to be a material constant.
- p.23, Fig. D1: The strain over time plots are only correct for a constant viscosity eta. These plots are confusing because in the paper the viscosity is always the nonlinear Glen-type viscosity.
  Furthermore, the authors should delete the dashed lines (as they are also confusing) and maybe give its explanation on the sides near the already existing arrows.
- p.23, l.7,8: The Kelvin model does also not capture the long-term viscous behavior and is a representation of a solid and not like the Maxwell material which represents a viscoelastic fluid. Rephrase it for a better understanding.
- p.24, l.14: The correlation between the bulk modulus K and the two elastic material parameters for an isotropic material, namely Young's modulus and Poisson's ratio, is missing. The same thing is missing for the shear modulus G. Is the viscosity the authors take for the Kelvin model not to small to see any influence? The order of the Maxwell viscosity from Glen's flow law is two orders of magnitude higher or is this not the case?

Technical corrections:
- p.1, l.1: "shows" (an extensive network is singular)
- p.1, l.10: "Filchner-Ronne Ice Shelf"
- p.1, l.11: delete the in "We evaluate all relevant mechanisms…"
- p.2, l.8,9: delete as such and in the following line a: "…in response to tidal forcing."
- p.2, l.27: a instead an "….using a viscous model."
- p.2, l. 11 and p.2, l.32 adjust notation of semi-diurnal in the whole manuscript
- Figure1: state the unit of tidal range in the caption or directly in the figure (dashed contour lines)
- Table 1: "Uses an exponent of 4…." instead of and
- p.6, l.7,11: abbreviation of three-dimensional is missing
- p.7, l.5 f.: (comma) definition of (.)…,… is missing
- p.7, l.8: definition of deviatoric stress and strain is missing
- p.7, l.8: a rheological model is the graphic depiction of spring and dashpot (cf. figure D1); delete rheological in "…an upper-convected Maxwell model…"
- p.7, l.11: definition or reference for the upper-convected time derivative is missing
- p.7, l.18: definition of the water density $rho_w$ is missing
- p.7, l.21: bold face of sigma as sigma is the stress tensor. Until now the authors only define its components
- p.8, l.8: CATS2008a instead of "Cats2008a", revise this in the whole paper
- p.10, l.12: outlet instead of "oulet"
- p.15, l.11: "is symmetric."
- p.20, l.14: unphysical without hyphen
- p.22, l.9: Did the authors mean $T\_B$ for the basal temperature instead of $T\_S$?
- p.23, l.2: viscoelastic instead of "viscoelsatic"
- p.24, l.15: modulus instead of "Modulus"
- p.24, l.26: the bracket should end two words later
- p.25, l.20: i.e.
- p.26, caption Fig. E1: a verb is missing in the last sentence
- p.26, l.5: Why is $U\_T$ boldface? It is just a number.

I enjoyed reading the paper. Best regards
Julia Christmann

---

## Referee Comment (RC2) · Anonymous Referee #2 · 26 Jun 2019

Sebastian H. R. Rosier and
G. Hilmar Gudmundsson

**Anonymous Referee #2**

In this study, the authors use a sophisticated 3D viscoelastic model of ice shelf flow to calculate the horizontal flow response of the Filchner-Ronne Ice Shelf (FRIS) to tidal forcing. The study is meant to be a comprehensive examination of the many theories that have been put forth to explain the large amplitude response of ice flow at FRIS at both $M_2$ and $M_{sf}$ tidal frequencies, all within a single modeling framework. In this goal, I think the authors have generally been successful, and this is an impressive achievement which brings together many ideas and observations spanning the last decade or so. What is perhaps most interesting, is how many theories (including

their own) the authors are able to rule out for explaining the conundrum of the large amplitude $M_{sf}$ signal at FRIS. This study provides what I think is a valuable "refocusing" of the conversation around low frequency tidal signals in ice shelf flow and how to explain/model them.

Though I am generally happy with the scientific content of this study, in what follows, I mainly point out some issues regarding information that is missing, information that could be condensed, and general textual edits that could be made to the manuscript.

Major points:

1. The manuscript has a narrative quality that takes the reader through the process of trial-and-error followed by the authors in eliminating possible explanations for the observations and coming to a final conclusion. This is fine and I think actually helpful to bring negative results into the literature in a way that is rarely done nowadays. That being said, the discussion in section 3.2 is often a bit cumbersome and overly detailed and will sometimes talk about the results of certain experiments (more on this in comments below) before they have been described. It would be worth it to condense the discussion on pages 11-13 a bit and spend more time talking about the details of the successful GLmigration experiments (which are currently a bit light on detail). I have a similar critique of the discussion and conclusion sections which re-hash many of the details right after they are discussed in section 3.2, these could also be streamlined to focus on what you learned from the successful simulations.

2. In all the figures (3-5) comparing modeled amplitudes to observed amplitudes, the reader must guess the correspondence by eye. This is not really feasible and doesn't permit an easy identification of spatial vs. constant-offset mismatches.

What would enable this comparison better is a scatter plot of the modeled amplitude vs. observed amplitude at the GPS sites for all simulations. This could be done by adding a single panel to each figure with markers for each experiment distinguished by color and/or marker type.

3. Overall, you do a nice job of constructing clean experiments that test the role of individual mechanisms in generating the tidal signal of interest. I think it is fair to say that you find the only mechanism which can really generate anything near the observed $M_{sf}$ amplitudes for reasonable parameters is GL migration. But then you go off on this exploration of how you could have slightly less GL migration if you also increase the Glen's law nonlinearity. This is true, but you could also slightly boost the amplitude using many of the other mechanisms that you tried as well. The point isn't necessarily that the other processes (including a high value of n) aren't happening, its just that they likely aren't playing much of a role in generating the $M_{sf}$ signal. I think the $n = 5$ experiment in Figure 5 muddies this point by selectively picking one of the "failed" mechanisms to bring back into the conversation. It would be better to leave it out (or to a supplement) I think.

4. When you describe the model in section 2, you specify certain choices (e.g., boundary conditions) which you later loosen or change. It would be helpful if you said up front, we set X for the default simulation, but we will loosen this assumption in later simulations. Just so that the reader is prepared to know what choices are constant across all experiments, and which are changed.

5. You do a nice job of reviewing the literature on proposed mechanisms for the FRIS $M_{sf}$ signal early in the manuscript. However, later when you discuss the experiments you try and which are successful there is less attribution of ideas to previous studies. In particular, the idea that asymmetries in grounding line migration can produce a large $M_{sf}$ signal through changing basal contact and pinning points was originally put forward in Robel et al. (2017). Yet there is no

discussion of what that study gets right and wrong in any of sections 3.2, 4 or 5. It would be helpful to discuss and evaluate these previous studies in more detail, as you have done for some of the other mechanisms discussed.

Minor points:
P. 1 Line 6: entire Filchner-Ronne
P. 1 Line 11: tidal ice flow response
P. 1 Line 5-11: this is a bit detailed for the abstract, perhaps compress this discussion somewhat
P. 1 Line 19: flows, on average, $21\%$
P. 1 Line 21: floating ice shelves
P. 1 Line 23: who is the "us" here? did you do this work?
P. 1 Line 24-25: Ice shelves are not thought to modify upstream flow, and also to have a fundamental effect on the....
P. 2 Line 4: delete sentence: "Ice shelves...viscous processes" - more confusing than illuminating
P. 2 Line 4-5: increasing quantity of GPS and InSAR observations
P. 2 Line 8: delete "while...as such"
P. 2 Line 12: ones -> frequencies
P. 2 Line 12: delete "long-periodic"
P. 2 Line 14: several times in the paper you state that a nonlinear mechanism is needed to transfer tidal energy from short to long periods. Why? Readers not versed in this literature (or the idea of frequency mixing) will not find this to be an obvious statement. Could be helped by a brief explanation here of heterodyning/freq mixing.
P. 2 Line 16: models that only include elastic flexure
P. 2 Line 17: delete "with exact numbers dependent on location"
P. 2 Line 23-24: explain "the mechanical coupling between vertical ocean tides and ice flow occurs in the grounding zone"
P. 2 Line 24: our modelling and observational efforts

P. 2 Line 27: using a viscous

P. 2 Line 31: will be to replicate

P. 2 Line 34: suggested in previous studies as responsible

Figure 1: there's about 5 different types of information on this map? Are all of them strictly necessary (i.e. is elevation)? Would be helpful if you simplified.

P. 4 Line 7: constituents have the largest amplitude in this region

P. 4 Line 12: motion, but not

P. 4 Line 14: constituents, these do not rotate around an amphidromic point, and instead

P. 4 Line 22: signal and the (horizontal) ice shelf flow response

P. 4 Line 23: in the Weddell Sea, the semidiurnal

P. 4 Line 29: tidal cycle, periodically causing direction

P. 4 Line 33: diurnal

P. 5 Line 22: also these studies have shown that the $M_{sf}$ signal occurs in the ice shelf first (using phase information)

P. 6 Line 4: processes using a X, Y, Z model

P. 6 Line 5: focus on modelling the two strongest responses observed in horizontal

P. 6 Line 13: angular momentum (respectively):

P. 7 Line 8: what does upper-convected mean in this context

P. 7 Line 5: explain the $i, j$ indices

P. 7 Line 14: from what observation is the rate factor inverted? using what inversion approach? I know the detail is in appendix, but summarize in one sentence here

P. 7 Line 15: what are the other parameters(a table would be helpful) and how are they set. You get at these details somewhat (regarding E) later, but would be more useful here.

P. 7 Line 19: At the ice surface, a stress-free boundary

P. 8 Line 1: are $u_{obs}$ and $v_{obs}$ constant in time?

P. 8 Line 3: here is one place where it would be helpful to explicitly state that vertical clamping at grounding effectively prevents GL migration in the default setup, but this assumption will be loosened later on to allow grounding line migration to occur

P. 8 Line 15: interpolation is used to

P. 8 Line 5: the other benefit of only including principal diurnal and semidiurnal constituents is that you ensure that the presence of any other constituents in the ice shelf response has to do with processes in the flow and not the forcing itself

P. 8 Line 24: how do you generate the mesh?

P. 8 Line 30: this implies that the mechanisms which produce $M_2$ and $M_{sf}$ flow variability are not the same. Perhaps it is worth saying this explicitly?

P. 9: Please provide more details on what you mean by "processing surface model displacements using the Tide MATLAB package"

Figure 3 and elsewhere: you often say things like "ice shelf motion" - this is really vague. Is this horizontal displacement? is it detrended? please be more specific on this

P. 10 Line 14: you refer to $RF_{streams}$ here before even describing what it is. Either do that here, or remove this reference

P. 10 Line 18: its it's

P. 11 Line 6-10: why was no $M_2$ signal generated in the grounding zone

P. 11 Line 13-15: delete "we start by…fig 4" sentence - or move down, it doesn't make sense here

P. 11 Line 20: rheology which is the source of nonlinearity in this mechanism

P. 11 Line 24-25: how can we tell the origin from the plots in figure 4? Are you relying on phase information that isn't plotted? It may be useful to say something about phasing.

Figure 4: the colorbar of observed amplitudes is inconsistent with the modeled colorbar (and pretty much every other figure). I can see that there will be a large discrepancy, but that's the point, isn't it? Also, plot observed filled circles on all

[Figure]

panels.

P. 13 Line 4: model assumes hydrostatic

P. 13 Line 30-35: you introduce the gammas, but don't explain how they are calculated so we have no context for what these numbers mean, or that they come from Tsai and Gudmundsson (I know its in the appendix, but since this mechanism is important for this study, its worth explaining in the main text)

P. 16 Line 2: exploring the processes and parameters that play a role in generating $M_{sf}$ signal

P. 16 Line 11: found that the amplitude of the $M_{sf}$ signal in observations

P. 16 Line 14: difference

P. 18 Line 5: tidal modulation

---

## Author Comment (AC1) · 23 Jul 2019

We are extremely grateful to Julia Christmann for going through the manuscript in detail and providing very helpful comments. In the text below we have addressed (in red italics) all the main points that were raised (in bold). All minor points/corrections will be implemented into a revised version of the manuscript if we are invited to submit one.

Sebastian Rosier and Hilmar Gudmundsson

**Anonymous Referee #1**

The paper "Exploring mechanisms responsible for tidal modulation in flow of the Filchner Ronne Ice Shelf" by Sebastian H. R. Rosier and G. Hilmar Gudmundsson presents an interesting comparison of GPS measurements and simulation results considering the influence of ocean tidal forcing. The authors explore the impact of different mechanisms on tidal modulation in horizontal flow. First, the modelled semi-diurnal tidal constituent M2 for the whole Filchner-Ronne Ice Shelf is compared to results of GPS measurements and afterwards the same is done for the long-periodic tidal constituent Msf. The authors claim that the relatively high amplitude for Msf can only be obtained if the model allows grounding line migration.

General comments:

I have a number of major concerns with this paper, which are detailed below. In addition, some more specific comments and technical corrections are listed at the end of this review.

- **The authors show in Table 1 an overview of the various model versions. Can you also include a Table where all parameters of the default setup are included, please? Then it is directly clear to the reader of the paper which parameter change is done for a different model setup and all parameter values are defined. For example, a value for Poisson's ratio is completely missing in the paper.**
  *We will add a table of the model parameters for the default setup to the revised manuscript, together with a description of the Poisson's ratio which is currently missing.*
- **For most model setups (except RF_streams), the velocity directly at the grounding line is set to an observed value but assumed to be constant – i.e. independent of the tidal forcing. In reality, the tides will have a big influence on the velocities near the grounding line, which is also verified by GPS measurements the authors show in this paper (shortly noticed p.10, l.13-15). The major simplification of a constant horizontal velocity is hardly discussed in the paper. Would the results be different if the horizontal velocity at the GL depends on tidal forcing? This could be checked by adding a sinusoidal behaviour onto the prescribed velocities at the boundary, with amplitudes and frequency derived from measurements. In Figure 4c (RF_streams), several ice streams are added but the results are only slightly better. However, how long are the added ice streams? At their boundary, the authors also prescribe constant velocities. So maybe it is impossible to model the influence of the tides by this extended geometry as the length of the grounded ice parts is too short to get independent horizontal velocities (of the inflow boundary condition) at the GL. Why can the authors do this major simplification and then state by the process of elimination that the strong long-periodic Msf component is only obtained by grounding line migration?**
  *We agree with the reviewer and this is why we included the 'RF_streams' experiment, we believe that it addresses these concerns. As the reviewer states, tides are known to influence velocities at and upstream of ice stream grounding lines. The 'RF_streams' experiment includes all major ice streams flowing into the domain. All ice streams in this experiment (apart from Moller which is*

*wide and slow flowing) extend over 100km upstream of the grounding line which is the maximum distance at which tidal perturbations in ice flow have been observed. Their lateral extent is determined by the shear margins, such that the lateral side walls of the ice streams are located where ice flow speed is approximately zero. As such, the fact that the ice stream boundary condition forces a constant velocity does not affect our results (for this experiment). We actually began our modelling work treating this setup as the 'default' but found that removing the ice streams had a negligible effect on the results (as we show in the paper). This is both interesting in of itself and distils the problem to a simpler one thus reducing the number of potential parameters. Forcing the model at its boundary with an Msf signal would cause the signal to be generated at the boundary but all the evidence points to this being a process generated within the ice shelf. This would not therefore shed any light on what causes the observed Msf signal.*

- **Figure 2 (c): Why are there such large excursions in the geometry of the Filchner Ronne Ice Shelf? For example directly at the ice shelf front of Filchner Ice Shelf or on the left and right side of Ronne Ice Shelf. Also at other places irregularities are noticeable.**
  **In the caption, the authors forgot to include (panel c).**
  *Depending on which specific features the reviewer is referring to these are ice rises and small outlet glaciers feeding into the shelf whose bathymetry is vertically exaggerated. We will add a reference to panel c in the revised manuscript.*

- **To better understand the effects of the various model setups shown in this paper, it would be nice to have relative deviations between observed and modelled results. For instance, in Fig. 4 the reader can imagine the influence of the different cases, but cannot really detect how a certain change in the setup reduces or increaes the difference between simulation and observation. The Filchner-Ronne Ice Shelf covers a large area and the GPS measurements are only points increased in size to be visible.**

- *Both reviewers have suggested improvements in how we could present our model results and we are grateful for these suggestions. We will implement a combination of these suggestions that works well and produce improved figures in the revised manuscript.*

- **Did the authors do a study considering the influence of the applied mesh to the results? From my experience, a Maxwell model as every viscoelastic material model needs a very very fine mesh resolution (I agree that it is not possible to get such a fine mesh resolution for the whole Filchner Ronne Ice Shelf) to generate reasonable results that are nearly mesh-independent. Is the mesh resolution the authors take not too coarse for a viscoelastic material model? Are the results reliable? An additional question about the mesh that remains unclear for the reader: How many layers are used for the vertical model coordinate?**
  *We tested mesh resolution by doubling the number of elements both horizontally and vertically and found that these did not significantly affect our results. We will add a brief description of these tests in the revised manuscript.*

- **p.15, l.20-29: Why is Young's Modulus not a material constant? In all tested setups in your paper Young's modulus is constant using E=2.4 GPa except in the damage setup where Young's modulus is spatially changed but not in time. Or did I understand this incorrectly? Why should Young's modulus be a function of loading frequency for a viscoelastic material? I also wonder why are the viscoelastic properties of ice shelves better provided by your simulation? And the last point I do not really understand is what do you mean by cumulative elastic strain, if you fit Young's modulus to GPS observations?**
  *Our wording here could be improved as we can see how this leads to confusion. The Young's Modulus as defined in the Maxwell model is a constant in most of our simulations and fixed at 2.4GPa. Our point stands however, that for a viscoelastic material subject to a periodic forcing, the concept of a constant elastic modulus breaks down and 'E' becomes a complex dynamic*

*modulus that is a function of forcing frequency. This is an important point to make since it has been overlooked in many previous studies of tidal behaviour.*

*Regarding the second point, as we show in the paper the horizontal motion at ice the front at semidiurnal frequencies is generated as an elastic response to the ice shelf tilting as the tides rotate around the Weddell Sea. Hence, the M2 signal we see in the model is not locally generated and is a result of cumulative elastic strain over the entire ice shelf. Thus, it does not provide a local estimate of elastic rheology (as is the case for all other previous experiments) but an integrated estimate over the entire ice shelf. We will try and make this point clearer in the revised manuscript.*

- **p.16, l.6-9: In which setup did the authors test periodic narrowing and widening of the ice shelf and reduced buttressing from ice rises/rumples?**
  *Adding grounding line migration to the model implicitly includes these two processes, since as the grounding line retreats/advances the ice shelf widens/narrows and ice rises within our domain unground/reground reducing/increasing buttressing.*

- **In my opinion, the conclusion would benefit from a reduction to clearly stated important take-home messages. Maybe the authors can move some of the statements from the conclusions into the discussion?**
  *The conclusion is admittedly rather long but there are quite a few results in the paper that need highlighting. The last paragraph of the conclusions is our best attempt to wrap everything up into the main take home messages. We will look to reduce the length of the conclusions where we can in a revised version of the manuscript.*

Specific comments and questions:

- **p.2, l.15,16: Give a reference. What is the expectation based on simple models of elastic flexure?**
  *We will add a reference to cover this in the revised manuscript.*
  **Are the numbers in the next line results the authors show later on?**
  *Yes, this will be rephrased for clarity.*

- **p.4, l.32: How long are the measurements of the additional GPS sites near the outlet glaciers?**
  *A large number of GPS sites are described in the cited paper and the measurement period varies from weeks to over a year.*

- **Table 1: Include an additional table that contains the material parameters for the default setup. Then it is clear which changes between the various model versions are done. What is the value of Poisson's ratio?**
  *This will be added to the revised manuscript. We use a Poisson's ratio of 0.41 in line with previous studies.*

- **What are the unknows of the model (velocities or displacements), i.e. do the authors use a velocity or displacement formulation for the viscoelastic material model? For the boundary conditions, the authors use Dirichlet conditions for the velocities (p.7, l.25) and also show the resulting ice velocity field in Fig. 2c, but for the element discretization the authors write (p.8, l.22) "triquadratic interpolation shape functions are used for displacements". Is an arbitrary Lagrangian–Eulerian moving grid included in the model or how are the surface nodal displacements (p.9, l.2) determined in the model?**
  *The unknowns are displacements, the Dirichlet boundary condition is implemented as a displacement divided by the time step.*

- **p.7, l.18: For the water pressure applied at the ice front one needs the freeboard also depending on the time-varying local sea level (see the if condition "z<0"). Is this only a typo or how do the authors implement this?**
  *The reviewer is correct, this should be written such that z is also a function of time and that is how it is implemented in the model, we will correct this in the revised manuscript.*

- **p.8, l.2-12: The reader cannot see the increase of the M2 amplitude towards the grounding zone. A second figure with a magnification near a chosen grounding line would be helpful.**
  *This will be added to the revised manuscript*

- **p.11, l.4-6: When the vertical boundary condition is removed, the grounding line has to move and in my view, bending will always occur near the real position of the grounding line but maybe not at the position where the grounding line was before. What do the authors mean with "removing the effect of bending in the grounding zone"?**
  *The point of this experiment is to remove the effect of bending stresses generated in the grounding zone. It is not a 'realistic' simulation but serves to shed light on what mechanisms are responsible for each part of the observed ice shelf response. The ice shelf does not bend in any meaningful way once this boundary condition is removed, perhaps the reviewer is referring to the tilting of the ice shelf?*

- **p.13, l.35: What happens for a positive or negative tidal motion of 4 m, which fits better to the tidal range given in Fig. 1 for the grounding line region?**
  *The migration distance increases linearly with tidal amplitude, and thus the Msf amplitude will also increase. However the 8m tidal range is only present in a limited part of the domain and it is highly unlikely that the steep sidewalls in this region allow such large migration distances so we don't see any benefit in including results for many different tidal ranges.*

- **p.21, l.7: Why did the authors put square brackets around K?**
  *These will be removed*

- **p.22, Fig. C1: The damage factor could reach a value of 0.8 and below in interesting regions, for example at the boundary of inflow regions to the ice shelf. In the text, the authors stated values of E between 1 and 9 GPa, but 0.2*2.4 GPa = 0.48 GPa. Are these realistic (meaning physically useful) values? In my opinion, E has to be a material constant.**
  *Firstly, these refer to different experiments. We tested a Young's Modulus of between 1 and 9 GPa to match the observed M2 signal on the ice shelf. We chose this range because most studies of elastic properties of glacial ice find E to lie within this range (although there are some considerably outside of it). In the damage experiment, the reviewer is correct that in some regions the high damage will lead to a very low 'effective young's modulus' – but that is in the nature of the continuum damage mechanics modelling approach. The aim is not to derive realistic values for E, the aim is to attempt to model fractured ice as a continuum by representing the effects of damage on the material stiffness.*

- **p.23, Fig. D1: The strain over time plots are only correct for a constant viscosity eta. These plots are confusing because in the paper the viscosity is always the nonlinear Glen-type viscosity. Furthermore, the authors should delete the dashed lines (as they are also confusing) and maybe give its explanation on the sides near the already existing arrows.**
  *We will redraw this figure to make it clearer and address these concerns.*

- **p.23, l.7,8: The Kelvin model does also not capture the long-term viscous behavior and is a representation of a solid and not like the Maxwell material which represents a viscoelastic fluid. Rephrase it for a better understanding.**
  *We do not state anywhere that the Kelvin model captures long-term viscous behaviour and do not understand the reviewers comment.*

- **p.24, l.14: The correlation between the bulk modulus K and the two elastic material parameters for an isotropic material, namely Young's modulus and Poisson's ratio, is missing. The same thing is missing for the shear modulus G. Is the viscosity the authors take for the Kelvin model not too small to see any influence? The order of the Maxwell viscosity from Glen's flow law is two orders of magnitude higher or is this not the case?**
  *We define the shear modulus G by E and nu in equation 5 and we are not sure what is to gain from defining K since we only show the deviatoric viscoelastic equation which does not include K. We would prefer to keep the details of the viscoelastic rheology to a minimum since is likely to confuse readers who may be unfamiliar with these relations. We define the viscosity of the Kelvin element in line with previous studies although we tested various viscosities and did not find a value within a sensible range that lead to an improved fit with observations.*

Technical corrections:

• p.1, l.1: "shows" (an extensive network is singular)

• p.1, l.10: "Filchner-Ronne Ice Shelf"

• p.1, l.11: delete the in "We evaluate all relevant mechanisms…"

• p.2, l.8,9: delete as such and in the following line a: "…in response to tidal forcing."

• p.2, l.27: a instead of an "….using a viscous model."

• p.2, l. 11 and p.2, l.32 adjust notation of semi-diurnal in the whole manuscript

• Figure1: state the unit of tidal range in the caption or directly in the figure (dashed contour lines)

• Table 1: "Uses an exponent of 4…." instead of and

• p.6, l.7,11: abbreviation of three-dimensional is missing

• p.7, l.5 f.: (comma) definition of (.)…,… is missing

• p.7, l.8: definition of deviatoric stress and strain is missing

• p.7, l.8: a rheological model is the graphic depiction of spring and dashpot (cf. figure D1); delete rheological in "…an upper-convected Maxwell model…"

• p.7, l.11: definition or reference for the upper-convected time derivative is missing

• p.7, l.18: definition of the water density rhow is missing

• p.7, l.21: bold face of sigma as sigma is the stress tensor. Until now the authors only define its components

• p.8, l.8: CATS2008a instead of "Cats2008a", revise this in the whole paper

• p.10, l.12: outlet instead of "oulet"

• p.15, l.11: "is symmetric."

• p.20, l.14: unphysical without hyphen

• p.22, l.9: Did the authors mean $T\_B$ for the basal temperature instead of $T\_S$?

• p.23, l.2: viscoelastic instead of "viscoelsatic"

- p.24, l.15: modulus instead of "Modulus"

- p.24, l.26: the bracket should end two words later

- p.25, l.20: i.e.

- p.26, caption Fig. E1: a verb is missing in the last sentence

- p.26, l.5: Why is $U\_T$ boldface? It is just a number

---

## Author Comment (AC2) · 23 Jul 2019

We are extremely grateful to the anonymous reviewer for going through the manuscript in detail and providing very helpful comments. In the text below we have addressed (in red italics) all the main points that were raised (in bold). All minor points/corrections will be implemented into a revised version of the manuscript if we are invited to submit one.

Sebastian Rosier and Hilmar Gudmundsson

**Anonymous Referee #2**

In this study, the authors use a sophisticated 3D viscoelastic model of ice shelf flow to calculate the horizontal flow response of the Filchner-Ronne Ice Shelf (FRIS) to tidal forcing. The study is meant to be a comprehensive examination of the many theories that have been put forth to explain the large amplitude response of ice flow at FRIS at both M2 and Msf tidal frequencies, all within a single modeling framework. In this goal, I think the authors have generally been successful, and this is an impressive achievement which brings together many ideas and observations spanning the last decade or so. What is perhaps most interesting, is how many theories (including their own) the authors are able to rule out for explaining the conundrum of the large amplitude Msf signal at FRIS. This study provides what I think is a valuable "refocusing" of the conversation around low frequency tidal signals in ice shelf flow and how to explain/model them. Though I am generally happy with the scientific content of this study, in what follows, I mainly point out some issues regarding information that is missing, information that could be condensed, and general textual edits that could be made to the manuscript.

Major points:

**1. The manuscript has a narrative quality that takes the reader through the process of trial-and-error followed by the authors in eliminating possible explanations for the observations and coming to a final conclusion. This is fine and I think actually helpful to bring negative results into the literature in a way that is rarely done nowadays. That being said, the discussion in section 3.2 is often a bit cumbersome and overly detailed and will sometimes talk about the results of certain experiments (more on this in comments below) before they have been described. It would be worth it to condense the discussion on pages 11-13 a bit and spend more time talking about the details of the successful GLmigration experiments (which are currently a bit light on detail). I have a similar critique of the discussion and conclusion sections which re-hash many of the details right after they are discussed in section 3.2, these could also be streamlined to focus on what you learned from the successful simulations.**

*The paper was somewhat difficult to write due to large number of different model experiments. We would argue the manuscript is more ponderous than cumbersome but we agree that there is room for improvement! We will take on board this advice and streamline the later sections of the paper to hopefully condense our discussion without missing important results.*

**2. In all the figures (3-5) comparing modeled amplitudes to observed amplitudes, the reader must guess the correspondence by eye. This is not really feasible and doesn't permit an easy identification of spatial vs. constant-offset mismatches. What would enable this comparison better is a scatter plot of the modeled amplitude vs. observed amplitude at the GPS sites for all simulations. This could be done by adding a single panel to each figure with markers for each experiment distinguished by color and/or marker type.**

*Both reviewers have suggested improvements in how we could present our model results and we are grateful for these suggestions. We will implement a combination of these suggestions that works well and produce improved figures in the revised manuscript.*

**3. Overall, you do a nice job of constructing clean experiments that test the role of individual mechanisms in generating the tidal signal of interest. I think it is fair to say that you find the only mechanism which can really generate anything near the observed Msf amplitudes for reasonable parameters is GL migration. But then you go off on this exploration of how you could have slightly less GL migration if you also increase the Glen's law nonlinearity. This is true, but you could also slightly boost the amplitude using many of the other mechanisms that you tried as well. The point isn't necessarily that the other processes (including a high value of n) aren't happening, its just that they likely aren't playing much of a role in generating the Msf signal. I think the n = 5 experiment in Figure 5 muddies this point by selectively picking one of the "failed" mechanisms to bring back into the conversation. It would be better to leave it out (or to a supplement) I think.**

*We can see the point that the reviewer makes here but we would still argue that exploring how the grounding line migration mechanism is affected by changing the Glen's law exponent is a valuable part of the manuscript. Certainly other mechanisms that were tested could also have reduced the amount of grounding line migration needed to match the signal but choosing a value of n=5 had the largest impact (by a considerable margin) on Msf amplitude after the grounding line migration mechanism. Given this, and that the value for n is almost universally and somewhat arbitrarily fixed at 3 despite being poorly constrained, we felt that including this additional experiment would be of greatest interest to the community. With more observations and modelling it might become possible to use this approach to obtain additional constrains on the value of n. We hope that by including a better explanation for our choice in the manuscript that the reviewer will agree that this is worth keeping.*

**4. When you describe the model in section 2, you specify certain choices (e.g., boundary conditions) which you later loosen or change. It would be helpful if you said up front, we set X for the default simulation, but we will loosen this assumption in later simulations. Just so that the reader is prepared to know what choices are constant across all experiments, and which are changed.**

*We will ensure that the revised manuscript makes it clearer which modelling choices are fixed and which are changed for our various experiments.*

**5. You do a nice job of reviewing the literature on proposed mechanisms for the FRIS Msf signal early in the manuscript. However, later when you discuss the experiments you try and which are successful there is less attribution of ideas to previous studies. In particular, the idea that asymmetries in grounding line migration can produce a large Msf signal through changing basal contact and pinning points was originally put forward in Robel et al. (2017). Yet there is no discussion of what that study gets right and wrong in any of sections 3.2, 4 or 5. It would be helpful to discuss and evaluate these previous studies in more detail, as you have done for some of the other mechanisms discussed.**

*On reflection we agree with the reviewer that the later sections of our paper would be greatly improved by adding more discussion regarding previous studies and we will expand on this in the revised manuscript.*

Minor points:

P. 1 Line 6: entire Filchner-Ronne

P. 1 Line 11: tidal ice flow response

P. 1 Line 5-11: this is a bit detailed for the abstract, perhaps compress this discussion somewhat

P. 1 Line 19: flows, on average, 21%

P. 1 Line 21: floating ice shelves

P. 1 Line 23: who is the "us" here? did you do this work?

P. 1 Line 24-25: Ice shelves are not thought to modify upstream flow, and also to have a fundamental effect on the. . ..

P. 2 Line 4: delete sentence: "Ice shelves. . .viscous processes" - more confusing than illuminating P. 2 Line 4-5: increasing quantity of GPS and InSAR observations

P. 2 Line 8: delete "while. . .as such"

P. 2 Line 12: ones -> frequencies

P. 2 Line 12: delete "long-periodic"

P. 2 Line 14: several times in the paper you state that a nonlinear mechanism is needed to transfer tidal energy from short to long periods. Why? Readers not versed in this literature (or the idea of frequency mixing) will not find this to be an obvious statement. Could be helped by a brief explanation here of heterodyning/freq mixing.

P. 2 Line 16: models that only include elastic flexure

P. 2 Line 17: delete "with exact numbers dependent on location"

P. 2 Line 23-24: explain "the mechanical coupling between vertical ocean tides and ice flow occurs in the grounding zone"

P. 2 Line 24: our modelling and observational efforts

P. 2 Line 27: using a viscous

P. 2 Line 31: will be to replicate

P. 2 Line 34: suggested in previous studies as responsible

Figure 1: there's about 5 different types of information on this map? Are all of them strictly necessary (i.e. is elevation)? Would be helpful if you simplified.

P. 4 Line 7: constituents have the largest amplitude in this region

P. 4 Line 12: motion, but not

P. 4 Line 14: constituents, these do not rotate around an amphidromic point, and instead

P. 4 Line 22: signal and the (horizontal) ice shelf flow response

P. 4 Line 23: in the Weddell Sea, the semidiurnal

P. 4 Line 29: tidal cycle, periodically causing direction

P. 4 Line 33: diurnal

P. 5 Line 22: also these studies have shown that the Msf signal occurs in the ice shelf first (using phase information)

P. 6 Line 4: processes using a X, Y, Z model

P. 6 Line 5: focus on modelling the two strongest responses observed in horizontal

P. 6 Line 13: angular momentum (respectively):

P. 7 Line 8: what does upper-convected mean in this context

P. 7 Line 5: explain the i, j indices

P. 7 Line 14: from what observation is the rate factor inverted? using what inversion approach? I know the detail is in appendix, but summarize in one sentence here

P. 7 Line 15: what are the other parameters(a table would be helpful) and how are they set. You get at these details somewhat (regarding E) later, but would be more useful here.

P. 7 Line 19: At the ice surface, a stress-free boundary

P. 8 Line 1: are uobs and vobs constant in time?

P. 8 Line 3: here is one place where it would be helpful to explicitly state that vertical clamping at grounding effectively prevents GL migration in the default setup, but this assumption will be loosened later on to allow grounding line migration to occur

P. 8 Line 15: interpolation is used to

P. 8 Line 5: the other benefit of only including principal diurnal and semidiurnal constituents is that you ensure that the presence of any other constituents in the ice shelf response has to do with processes in the flow and not the forcing itself

P. 8 Line 24: how do you generate the mesh?

P. 8 Line 30: this implies that the mechanisms which produce M2 and Msf flow variability are not the same. Perhaps it is worth saying this explicitly?

P. 9: Please provide more details on what you mean by "processing surface model displacements using the Tide MATLAB package"

Figure 3 and elsewhere: you often say things like "ice shelf motion" - this is really vague. Is this horizontal displacement? is it detrended? please be more specific on this

P. 10 Line 14: you refer to RFstreams here before even describing what it is. Either do that here, or remove this reference

P. 10 Line 18: its it's

P. 11 Line 6-10: why was no M2 signal generated in the grounding zone

P. 11 Line 13-15: delete "we start by. . .fig 4" sentence - or move down, it doesn't make sense here

P. 11 Line 20: rheology which is the source of nonlinearity in this mechanism

P. 11 Line 24-25: how can we tell the origin from the plots in figure 4? Are you relying on phase information that isn't plotted? It may be useful to say something about phasing.

Figure 4: the colorbar of observed amplitudes is inconsistent with the modeled colorbar (and pretty much every other figure). I can see that there will be a large discrepancy, but that's the point, isn't it? Also, plot observed filled circles on all panels.

P. 13 Line 4: model assumes hydrostatic

P. 13 Line 30-35: you introduce the gammas, but don't explain how they are calculated so we have no context for what these numbers mean, or that they come from Tsai and Gudmundsson (I know its in the appendix, but since this mechanism is important for this study, its worth explaining in the main text)

P. 16 Line 2: exploring the processes and parameters that play a role in generating Msf signal

P. 16 Line 11: found that the amplitude of the Msf signal in observations

P. 16 Line 14: difference

P. 18 Line 5: tidal modulation

---

## Referee Report (RR1)

The paper "Exploring mechanisms responsible for tidal modulation in flow of the Filchner-Ronne Ice Shelf" by Sebastian H. R. Rosier and G. Hilmar Gudmundsson has improved a lot in the revision. I have only minor comments left. The comments highlighted in green are answers to the comments of the authors (highlighted in red). The technical corrections refer to the marked-up version of the manuscript (tc-2019-79-author_response-version.pdf).

Specific comments and questions:

- p.15, l.20-29: Why is Young's Modulus not a material constant? In all tested setups in your paper Young's modulus is constant using E=2.4 GPa except in the damage setup where Young's modulus is spatially changed but not in time. Or did I understand this incorrectly? Why should Young's modulus be a function of loading frequency for a viscoelastic material? I also wonder why are the viscoelastic properties of ice shelves better provided by your simulation? And the last point I do not really understand is what do you mean by cumulative elastic strain, if you fit Young's modulus to GPS observations?

  Our wording here could be improved as we can see how this leads to confusion. The Young's Modulus as defined in the Maxwell model is a constant in most of our simulations and fixed at 2.4GPa. Our point stands however, that for a viscoelastic material subject to a periodic forcing, the concept of a constant elastic modulus breaks down and 'E' becomes a complex dynamic modulus that is a function of forcing frequency. This is an important point to make since it has been overlooked in many previous studies of tidal behaviour.

  Regarding the second point, as we show in the paper the horizontal motion at ice the front at semidiurnal frequencies is generated as an elastic response to the ice shelf tilting as the tides rotate around the Weddell Sea. Hence, the M2 signal we see in the model is not locally generated and is a result of cumulative elastic strain over the entire ice shelf. Thus, it does not provide a local estimate of elastic rheology (as is the case for all other previous experiments) but an integrated estimate over the entire ice shelf. We will try and make this point clearer in the revised manuscript.

  We have reworded this paragraph so that both of these points are hopefully much clearer.

  I agree with the authors that for a viscoelastic material 'E' becomes a complex dynamic modulus for a periodic forcing with a sufficiently large frequency. But the frequency of the tidal forcing in ice is very low (<1e-4 1/s) and the response of the viscoelastic system will reduce to the one for a static load. Therefore, in my view, Young's modulus should be constant also for studies that model tidal behaviour.

- What are the unknows of the model (velocities or displacements), i.e. do the authors use a velocity or displacement formulation for the viscoelastic material model? For the boundary conditions, the authors use Dirichlet conditions for the velocities (p.7, l.25) and also show the resulting ice velocity field in Fig. 2c, but for the element discretization the authors write (p.8, l.22) "triquadratic interpolation shape functions are used for displacements". Is an arbitrary Lagrangian–Eulerian moving grid included in the model or how are the surface nodal displacements (p.9, l.2) determined in the model?

  The unknowns are displacements, the Dirichlet boundary condition is implemented as a displacement divided by the time step.

  Is the time step constant? How did the authors choose it? How long is it?

- p.11, l.4-6: When the vertical boundary condition is removed, the grounding line has to move and in my view, bending will always occur near the real position of the grounding line but maybe not at the position where the grounding line was before. What do the authors mean with "removing the effect of bending in the grounding zone"?

  The point of this experiment is to remove the effect of bending stresses generated in the grounding zone. It is not a 'realistic' simulation but serves to shed light on what mechanisms are responsible for each part of the observed ice shelf response. The ice shelf does not bend in any meaningful way once this boundary condition is removed, perhaps the reviewer is referring to the tilting of the ice shelf?

  The sentence has been reworded slightly

  The authors are totally right to test the role of all mechanisms represented in the model. Nevertheless, I suggest to mention somewhere that not all simulations are

designed to faithfully reproduce processes that occur in the real world. For instance, the experiment with n=5 is a (useful) sensitivity study rather than a 'realistic' simulation. Maybe the authors can shortly discuss which of the experiments are more realistic than others.

- p.13, l.35: What happens for a positive or negative tidal motion of 4 m, which fits better to the tidal range given in Fig. 1 for the grounding line region?
  The migration distance increases linearly with tidal amplitude, and thus the Msf amplitude will also increase. However the 8m tidal range is only present in a limited part of the domain and it is highly unlikely that the steep sidewalls in this region allow such large migration distances so we don't see any benefit in including results for many different tidal ranges.
  But the tidal range of 8m occurs very near the grounding line (Fig.1). I would expect that the migration of the grounding line is mainly influenced by the tidal amplitudes nearby. Is this not the case?

- p.22, Fig. C1: The damage factor could reach a value of 0.8 and below in interesting regions, for example at the boundary of inflow regions to the ice shelf. In the text, the authors stated values of E between 1 and 9 GPa, but 0.2*2.4 GPa = 0.48 GPa. Are these realistic (meaning physically useful) values? In my opinion, E has to be a material constant.
  Firstly, these refer to different experiments. We tested a Young's Modulus of between 1 and 9 GPa to match the observed M2 signal on the ice shelf. We chose this range because most studies of elastic properties of glacial ice find E to lie within this range (although there are some considerably outside of it). In the damage experiment, the reviewer is correct that in some regions the high damage will lead to a very low 'effective young's modulus' – but that is in the nature of the continuum damage mechanics modelling approach. The aim is not to derive realistic values for E, the aim is to attempt to model fractured ice as a continuum by representing the effects of damage on the material stiffness.
  Yes, sure the authors are right that a small effective Young's modulus can model highly fractured ice. But my question to the damage experiment is: Is the Filchner-Ronne Ice Shelf in the regions where the authors get high damage factors highly crevassed/damaged? See for example Filchner Ice Shelf where the mean flow velocities are pretty small. In these regions, the resolution of the mesh is additionally very coarse.

Technical corrections:
- p.16, l.17: leads to an increase of the $M_{sf}$ amplitude
- p.29, Figure E1: \delta S$^+$ instead of \delta h$^+$

Best regards
Julia Christmann

---

## Author Response (AR2)

We are very grateful to Julia Christmann for going through the manuscript again and providing her feedback. The document is colour coded as follows: black bullet points are the original reviewer comments, red text is our original reply, green text is the most recent reviewer comments and blue italicised text is our reply to these latest comments.

Sebastian Rosier and Hilmar Gudmundsson

Specific comments and questions:

• p.15, l.20-29: Why is Young's Modulus not a material constant? In all tested setups in your paper Young's modulus is constant using E=2.4 GPa except in the damage setup where Young's modulus is spatially changed but not in time. Or did I understand this incorrectly? Why should Young's modulus be a function of loading frequency for a viscoelastic material? I also wonder why are the viscoelastic properties of ice shelves better provided by your simulation? And the last point I do not really understand is what do you mean by cumulative elastic strain, if you fit Young's modulus to GPS observations?

Our wording here could be improved as we can see how this leads to confusion. The Young's Modulus as defined in the Maxwell model is a constant in most of our simulations and fixed at 2.4GPa. Our point stands however, that for a viscoelastic material subject to a periodic forcing, the concept of a constant elastic modulus breaks down and 'E' becomes a complex dynamic modulus that is a function of forcing frequency. This is an important point to make since it has been overlooked in many previous studies of tidal behaviour. Regarding the second point, as we show in the paper the horizontal motion at ice the front at semidiurnal frequencies is generated as an elastic response to the ice shelf tilting as the tides rotate around the Weddell Sea. Hence, the M2 signal we see in the model is not locally generated and is a result of cumulative elastic strain over the entire ice shelf. Thus, it does not provide a local estimate of elastic rheology (as is the case for all other previous experiments) but an integrated estimate over the entire ice shelf. We will try and make this point clearer in the revised manuscript.

We have reworded this paragraph so that both of these points are hopefully much clearer.

I agree with the authors that for a viscoelastic material 'E' becomes a complex dynamic modulus for a periodic forcing with a sufficiently large frequency. But the frequency of the tidal forcing in ice is very low (<1e-4 1/s) and the response of the viscoelastic system will reduce to the one for a static load. Therefore, in my view, Young's modulus should be constant also for studies that model tidal behaviour.

*The tidal frequency falls within the range for which both elastic and viscous effects are important which is why a viscoelastic model is essential. As a result, the system does not reduce to that of a static load and the Young's modulus is altered to reflect this. If the tidal frequency was significantly higher then we agree with the reviewer and a 'true' instantaneous value of the Young's modulus as measured by laboratory measurements (i.e. approximately 9GPa) would be appropriate, but in this case the Young's modulus must be altered to account for viscous effects.*

• What are the unknowns of the model (velocities or displacements), i.e. do the authors use a velocity or displacement formulation for the viscoelastic material model? For the boundary conditions, the authors use Dirichlet conditions for the velocities (p.7, l.25) and also show the resulting ice velocity field in Fig. 2c, but for the element discretization the authors write (p.8, l.22) "triquadratic interpolation shape functions are used for displacements". Is an arbitrary Lagrangian–

Eulerian moving grid included in the model or how are the surface nodal displacements (p.9, l.2) determined in the model?

The unknowns are displacements, the Dirichlet boundary condition is implemented as a displacement divided by the time step.

Is the time step constant? How did the authors choose it? How long is it?

*The time step changes adaptively based on a target number of iterations to reach certain specified convergence criteria. Furthermore, the nonlinear rheology is solved implicitly such that time stepping is much less of an issue than it would normally be.*

• p.11, l.4-6: When the vertical boundary condition is removed, the grounding line has to move and in my view, bending will always occur near the real position of the grounding line but maybe not at the position where the grounding line was before. What do the authors mean with "removing the effect of bending in the grounding zone"?

The point of this experiment is to remove the effect of bending stresses generated in the grounding zone. It is not a 'realistic' simulation but serves to shed light on what mechanisms are responsible for each part of the observed ice shelf response. The ice shelf does not bend in any meaningful way once this boundary condition is removed, perhaps the reviewer is referring to the tilting of the ice shelf?

The sentence has been reworded slightly

The authors are totally right to test the role of all mechanisms represented in the model. Nevertheless, I suggest to mention somewhere that not all simulations are designed to faithfully reproduce processes that occur in the real world. For instance, the experiment with n=5 is a (useful) sensitivity study rather than a 'realistic' simulation. Maybe the authors can shortly discuss which of the experiments are more realistic than others.

*We have altered the text to make it clear that both experiments described in this paragraph are purely to test the model and not designed to replicate reality. The results from both of these 'unrealistic' experiments are not discussed anywhere else or shown in any figures for this reason. With regards to changing the flow law exponent we do not agree with the reviewer that this is necessarily unrealistic; this value is a major source of uncertainty in ice sheet simulations and its appropriate value remains an open question.*

• p.13, l.35: What happens for a positive or negative tidal motion of 4 m, which fits better to the tidal range given in Fig. 1 for the grounding line region?

The migration distance increases linearly with tidal amplitude, and thus the Msf amplitude will also increase. However the 8m tidal range is only present in a limited part of the domain and it is highly unlikely that the steep sidewalls in this region allow such large migration distances so we don't see any benefit in including results for many different tidal ranges.

But the tidal range of 8m occurs very near the grounding line (Fig.1). I would expect that the migration of the grounding line is mainly influenced by the tidal amplitudes nearby. Is this not the case?

*Indeed the tidal amplitude at the grounding line is the important thing when considering grounding line migration. Possibly there is a misunderstanding about what we mean in this sentence: it is not that our parameters are specifically chosen to replicate a tidal motion of 3m.*

*This sentence is only to give some meaning to the parameters in terms of distance and amplitude which are more easily understood than slopes. The tidal amplitude in these experiments is not fixed to 3m, we use the actual tidal amplitude to force our model which will include the highest amplitudes of 4m near some of the grounding lines. We have re-worded this sentence to hopefully make this clearer.*

• p.22, Fig. C1: The damage factor could reach a value of 0.8 and below in interesting regions, for example at the boundary of inflow regions to the ice shelf. In the text, the authors stated values of E between 1 and 9 GPa, but 0.2*2.4 GPa = 0.48 GPa. Are these realistic (meaning physically useful) values? In my opinion, E has to be a material constant.

Firstly, these refer to different experiments. We tested a Young's Modulus of between 1 and 9 GPa to match the observed M2 signal on the ice shelf. We chose this range because most studies of elastic properties of glacial ice find E to lie within this range (although there are some considerably outside of it). In the damage experiment, the reviewer is correct that in some regions the high damage will lead to a very low 'effective young's modulus' – but that is in the nature of the continuum damage mechanics modelling approach. The aim is not to derive realistic values for E, the aim is to attempt to model fractured ice as a continuum by representing the effects of damage on the material stiffness.

Yes, sure the authors are right that a small effective Young's modulus can model highly fractured ice. But my question to the damage experiment is: Is the FilchnerRonne Ice Shelf in the regions where the authors get high damage factors highly crevassed/damaged? See for example Filchner Ice Shelf where the mean flow velocities are pretty small. In these regions, the resolution of the mesh is additionally very coarse.

*Overall our aim with the damage experiment is not to replicate the details of damage across the entire ice shelf but to introduce a plausible damage field that can at least serve to indicate whether or not including damage could explain the discrepancy between our model and observations. As such, many areas with damage do not necessarily correspond to crevassed regions of the ice shelf. In general, however, and particularly where mesh resolution is higher, damage is high in shear margins which is where crevasses would be expected to form on the ice shelf. The Mesh resolution is coarse in regions where mean flow velocities are small because the Msf signal is not generated in these areas. Thus whether or not the damage field is correct in these slow flowing regions should not affect our results.*

Technical corrections:

• p.16, l.17: leads to an increase of the Msf amplitude

*Done*

• p.29, Figure E1: \delta S+ instead of \delta h

*Done*

---

## Editor Decision (ED2)

[revised manuscript text omitted]

---

## Author Response (AR3)

Dear Evgeny Podolskiy,

Thank you very much for going through the manuscript in such detail, which revealed a number of minor mistakes that have all hopefully been corrected. We have implemented every one of your comments in the new revised version of the manuscript. The only point that perhaps deserves particular mention is regarding the 'xps' and 'yps' figure axes that you requested clarification on. These are shorthand for x/y polar stereographic but rather than explaining this detail in the paper we opted with an approach used by many other published studies using a similar system of replacing axes labels with a scale bar. We felt that the x/y coordinates do not help inform the reader in any way apart from giving a sense of scale and so hopefully the figures are now easier to interpret in this regard. Thank you once again for your work for this manuscript.

Kind regards,

Sebastian Rosier and Hilmar Gudmundsson